# NarrativeBridge: Enhancing Video Captioning with Causal-Temporal Narrative

**Asmar Nadeem**[1], **Faegheh Sardari**[1], **Robert Dawes**[2], **Syed Sameed Husain**[1],
**Adrian Hilton**[1], **Armin Mustafa**[1]

[1]CVSSP, University of Surrey, Guildford, UK
[2]BBC Research and Development, UK

{asmar.nadeem, armin.mustafa}@surrey.ac.uk

## Abstract

Existing video captioning benchmarks and models lack causal-temporal narrative, which is sequences of events linked through cause and effect, unfolding over time and driven by characters or agents. This lack of narrative restricts models' ability to generate text descriptions that capture the causal and temporal dynamics inherent in video content. To address this gap, we propose NarrativeBridge, an approach comprising of: (1) a novel Causal-Temporal Narrative (CTN) captions benchmark dataset generated using a large language model and few-shot prompting, explicitly encoding cause-effect temporal relationships in video descriptions; and (2) a Cause-Effect Network (CEN) with separate encoders for capturing cause and effect dynamics, enabling effective learning and generation of captions with causal-temporal narrative. Extensive experiments demonstrate that CEN significantly outperforms state-of-the-art models in articulating the causal and temporal aspects of video content: 17.88 and 17.44 CIDEr on the MSVD-CTN and MSRVTT-CTN datasets, respectively. Cross-dataset evaluations further showcase CEN's strong generalization capabilities. The proposed framework understands and generates nuanced text descriptions with intricate causal-temporal narrative structures present in videos, addressing a critical limitation in video captioning. For project details, visit https://narrativebridge.github.io/.

## 1 Introduction

Video captioning aims to generate textual descriptions that capture the visual information and temporal dynamics in videos Venugopalan et al. (2015). Research has primarily focused on developing new methods to improve the accuracy of video captioning models on well-established benchmark datasets, MSR-VTT Xu et al. (2016) and MSVD Chen & Dolan (2011). State-of-the-art (SOTA) approaches Iashin & Rahtu (2020); Tian et al. (2019); Xu et al. (2017); Nadeem et al. (2023); Wang et al. (2022); Chen et al. (2023) proposed new architectures to better align the generated captions with the provided ground-truth. However, these efforts enhance models' accuracy on existing evaluation criteria, without modifying the benchmark datasets or their ground truth annotations to address the lack of coherent representations of causal-temporal narrative. Causal-temporal narrative is the construction and interpretation of a sequence of events linked through cause and effect, unfolding over time and space, and often driven by entities (characters or agents) acting with intention Wilkens et al. (2003).

As discussed by Wilkens et al. (2003), causal relationships Granger (1969) are scenarios where one event (cause) directly influences the occurrence of another event (effect), while temporal understanding Berrevoets et al. (2023) involves recognizing the chronological order of events, a crucial component in establishing causality. Existing ground-truth captions in popular benchmarks lack causal, temporal and narrative information. Figure 1 illustrates the importance of causal-temporal narrative on the MSR-VTT dataset Xu et al. (2016). In the input video sequence, a car is first shown driving recklessly through an open field and flipping over, which represents the cause event. This is then followed by the effect event showing the severely damaged car, and subsequently, a group of guys starting to play beer pong, which represents the temporal sequence. The causal-temporal narrative would connect these events in a temporal order, linking the reckless driving and resulting car crash to the group's subsequent behavior of playing beer pong. This example aligns with the definition of narrative provided by Wilkens et al. Wilkens et al. (2003), which emphasizes the role of cause-effect relationships, the perception of narrativity in videos, and highlights the temporal

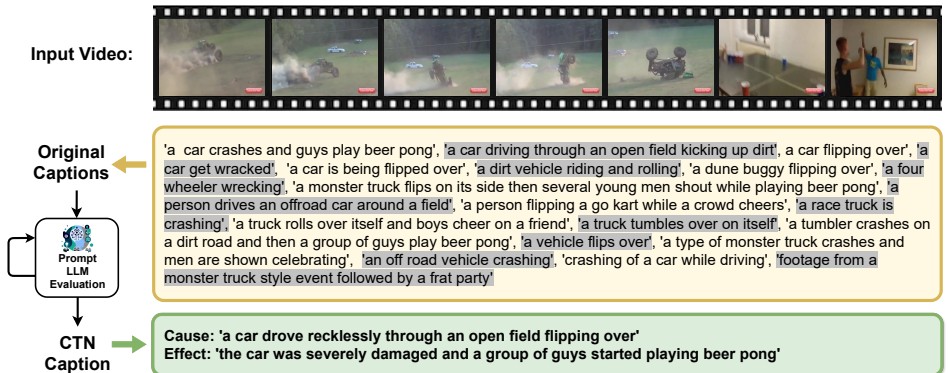

Figure 1: Comparison of Original captions vs. Causal-Temporal Narrative (CTN) caption to illustrate the inclusion of causal-temporal narrative.

sequence of events that is crucial for understanding the causal-temporal narrative. Figure 1 highlights the limitations of the original captions in existing benchmark datasets, such as MSR-VTT Xu et al. (2016). The original captions focus on isolated events or actions, such as "a car flipping over" or "a monster truck flips on its side," lacking contextual narrative and causal-temporal relationships between events. Consequently, models trained on these captions suffer from the same limitations.

To bridge this gap, we introduce NarrativeBridge, a novel framework encompassing a new benchmark dataset and architecture tailored for causal-temporal narrative learning in video captioning. Our Causal-Temporal Narrative (CTN), a novel captions benchmark dataset, leverages a large language model (LLM) and few-shot prompting to generate enhanced video descriptions that explicitly encode causal and temporal sequences, as shown in Figure 1. This establishes a clear connection between the cause (reckless driving) and the effect (damaged car and subsequent behavior of the group). Our CTN captions benchmark dataset enables models to better understand and articulate the causality, sequence, and significance of events within the broader video context Wilkens et al. (2003). To ensure the quality and relevance of the generated captions, we employ an automatic evaluation framework that compares the CTN captions with the video content, keeping or discarding them based on a score threshold. Additionally, we conduct a human evaluation study that further validates the high quality of our CTN captions, demonstrating their accuracy, temporal coherence, and relevance. This CTN captions benchmark dataset addresses the limitations of existing benchmark datasets and emphasizes the importance of incorporating causal-temporal narrative understanding into video captioning models to generate accurate, informative, and contextually relevant descriptions.

Existing SOTA video captioning methods that use different architectures such as LSTM Nadeem et al. (2023), GNN Hendria et al. (2023), and Transformer Wang et al. (2022), struggle to effectively learn causal-temporal narrative from the CTN captions. These architectures are designed to capture the overall semantics in videos but lack dedicated mechanisms to explicitly model the cause-effect relationships and temporal sequences. As a result, the captions generated by these methods fail to articulate the complex causal-temporal dynamics in the videos, as demonstrated in results (Section 4.3). To address this challenge, we propose the Cause-Effect Network (CEN) that captures cause and effect dynamics using dedicated encoders. By separately encoding the entire video through the cause and effect encoders, without requiring explicit cause-effect segmentation, and then combining them to generate the final caption, CEN is able to better understand the causal-temporal narrative. The primary contributions of our work are:

- Introducing for the first time CTN captions benchmark dataset with causal-temporal narrative captions, automatically evaluated and humanly validated.
- A novel CEN network for learning causal-temporal narrative information from the videos explicitly.
- Extensive experiments demonstrating CEN's superior performance on CTN, setting a new SOTA in causal-temporal narrative learning.

In addition, our work demonstrates strong generalization capabilities through cross-dataset evaluations and outperforms even fine-tuned SOTA vision-language models such as VideoLLaVA Lin et al. (2023) and ShareGPT4Video Chen et al. (2024) in generating causal-temporal narratives. Our work marks a significant step forward in video captioning research by explicitly addressing the challenges of causal-temporal narrative understanding. The CTN captions benchmark dataset and the CEN architecture provide a comprehensive framework for learning and generating narrative-based video descriptions.

## 2 RELATED WORK

### 2.1 BENCHMARKS

MSVD Chen & Dolan (2011) is a benchmark focused on human activities, providing a platform for evaluating video captioning models. The captions in MSVD often describe the observable actions without delving into the underlying motivations or the cause-effect relationships between the events. MSR-VTT Xu et al. (2016) is a large-scale benchmark with diverse video content, encompassing a wide range of topics and genres. The captions often focus on describing the observable content without capturing the causal links between the events or the temporal progression of the narrative. As a result, models trained on MSVD and MSR-VTT may struggle to generate descriptions that accurately reflect the causal and temporal dynamics in the videos.

While recent benchmarks like NExT-QA Xiao et al. (2021) and EgoSchema Mangalam et al. (2024) have made strides in incorporating causal and temporal reasoning in video understanding, they focus primarily on question-answering tasks rather than generating comprehensive causal-temporal narratives. NExT-QA introduces multi-choice and open-ended question-answering tasks focusing on specific question-answer pairs that often target single events or actions. In contrast, our CTN captions provide a more comprehensive narrative that captures the causal and temporal relationships across the entire video sequence (see Appendix A.7 for a detailed comparison). EgoSchema Mangalam et al. (2024), on the other hand, emphasizes long-form video understanding and temporal reasoning but does not explicitly focus on causal-temporal narrative.

Similarly, efforts like VCR Zellers et al. (2019), V2C Fang et al. (2020), and Motivation Vondrick et al. (2016) integrate causality into their analysis of visual description or question-answering, relying heavily on commonsense reasoning for generating predictions. VCR Zellers et al. (2019) focuses on visual commonsense reasoning, V2C Fang et al. (2020) aims to generate commonsense descriptions for video captioning, and Motivation Vondrick et al. (2016) explores the prediction of motivations behind actions in videos. However, these works primarily rely on commonsense reasoning and do not delve into the causal and temporal structures underpinning video narratives. Our CTN goes beyond existing benchmarks by explicitly modeling causal-temporal narrative in a single, coherent caption, enabling a more comprehensive understanding of video content.

### 2.2 VIDEO CAPTIONING

Video captioning techniques have evolved from LSTM-based Gao et al. (2017); Song et al. (2017); Nadeem et al. (2023) frameworks to the latest designs using SOTA GNNs Hendria et al. (2023); Zhang et al. (2020); Pan et al. (2020) and Transformers Wang et al. (2022); Lin et al. (2022); Yang et al. (2023), with a focus on enhancing the complexity of captions through the injection of multimodal data. Despite these advancements, current architectures often struggle to capture the intricate temporal sequences and causal relationships in video storytelling. To bridge this gap, video captioning can benefit from cross-fertilization with ideas and strategies developed in related fields, such as action recognition Sun et al. (2022); Wang et al. (2016); Kazakos et al. (2019); Xiao et al. (2020); Chen & Ho (2022); Gao et al. (2020); Panda et al. (2021); Sardari et al. (2023); Alfasly et al. (2022); Awan et al. (2024); Planamente et al. (2021); Zhang et al. (2022), event localization Tian et al. (2018); Lin et al. (2019); Duan et al. (2021); Lin et al. (2021), and question-answering Alamri et al. (2019); Hori et al. (2019); Schwartz et al. (2019); Geng et al. (2021); Yun et al. (2021); Li et al. (2022a); Shah et al. (2022); Nadeem et al. (2024). The integration of causal reasoning Liu et al. (2022); Xue et al. (2023) has shown promise in enhancing the ability of neural networks to discern causal relationships, leading to improved performance in image captioning Liu et al. (2022) and visual question answering Xue et al. (2023). However, current SOTA models still struggle to effectively handle the narrative complexity in videos.

Recent advancements in vision-language models (VLMs) such as VideoLLaVA Lin et al. (2023) and ShareGPT4Video Chen et al. (2024) have shown promising results in various video understanding tasks. However, as our experiments show (see Section 4.3), even these advanced models struggle with generating accurate causal-temporal narratives. In light of these challenges, our work explicitly addresses the limitations of the current approaches and provides a platform for causal-temporal narrative learning by introducing NarrativeBridge, a comprehensive framework that encompasses the CTN benchmark dataset and the CEN architecture.

## 3 METHOD

NarrativeBridge addresses the limitations of existing benchmarks and models in capturing causal-temporal narrative. It consists of two key components: (i) the CTN captions benchmark, which

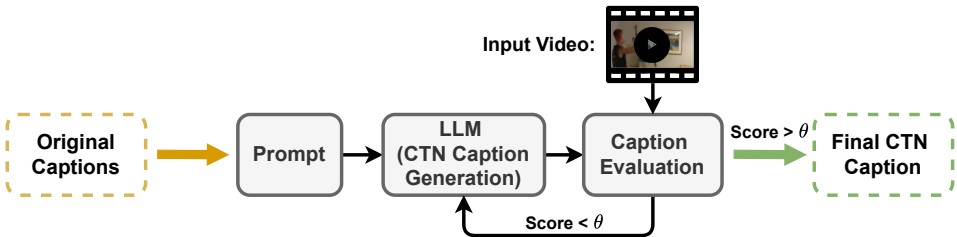

Figure 2: CTN caption generation pipeline. $\theta$ indicates a threshold.

provides a rich representation of cause-and-effect relationships and event sequences in video content, and (ii) the CEN architecture, designed to learn and articulate these causal-temporal narrative elements.

**Problem Statement:** Existing video captioning benchmarks lack the intricate causal-temporal narrative inherent in video content. SOTA video captioning methods also fall short in articulating the cause-and-effect relationships and temporal sequences that drive the events in the video (see Section 4.3).

## 3.1 CTN CAPTIONS BENCHMARK DATASET

To address the challenges of existing benchmarks in representing causal and temporal relationships within video content, we propose an approach that harnesses the potential of LLMs through the few-shot prompting technique Brown et al. (2020). Our method generates CTN captions without the need for model fine-tuning, leveraging the inherent generative capabilities of LLMs to produce captions that encapsulate causal-temporal narrative structures. Figure 2 shows the CTN caption generation pipeline, which consists of two key steps: prompt design, LLM-based caption generation and evaluation.

```
You are an advanced language model tasked with generating causal-temporal narrative captions for a
video. However, you cannot directly access the video itself. Instead, you will be provided with a
series of captions that outline the key events and scenes in the video. Your task is to generate a
concise Cause and Effect scenario, based on the information provided in the descriptive captions.
Be careful, your generated Cause and Effect statements should fulfill the following requirements:
1. Your narrative should be grounded in the information provided by the descriptive captions.
2. Cause and Effect scenario is relevant.
3. It should not introduce any new events or details not mentioned.
4. Avoid implying conclusions.
5. Maintain temporal consistency with the provided captions.
6. Use plain English and direct sentences.
7. Cause and Effect statements each limited to a maximum of 15 words.
8. Do not include any additional text before or after the JSON object.
Here are the examples of Cause and Effect:
[Examples]:
[{'Cause': 'the student overslept due to a malfunctioning alarm clock', 'Effect': 'missed catching the
bus to school'}, {'Cause': 'she absentmindedly skipped applying moisturizer after taking a long hot
shower', 'Effect': 'her skin became dry and flaky'}, {'Cause': 'he carelessly neglected taking his
prescribed allergy medication', 'Effect': 'suffered a severe sneezing fit'}, {'Cause': 'the exhausted
soccer player recklessly fouled an opponent in the penalty area', 'Effect': 'the opposing team was
awarded a crucial penalty kick'}, {'Cause': 'due to unforeseen road closures they found themselves
stuck in heavy traffic'', 'Effect': 'missed out on experiencing the opening act of the concert'}]
Now please generate only one Cause and Effect presented in a JSON format based on the following
descriptive captions.
[Descriptive Captions]:
<descriptive_captions>
[Causal Temporal Narrative]:
```

Prompt 1: LLM Prompt used in our CTN captions generation.

**Prompt Design:** The prompt design step is crucial in guiding the LLM to generate causal-temporal narrative captions. The input to the CTN caption generation are the original video captions from existing benchmarks. We design a prompt that include a small set of carefully selected example captions, illustrating the desired output structure and highlighting the key aspects of causal-temporal narrative. We design Prompt 1 which guides LLM to generate causal-temporal narrative. The prompt

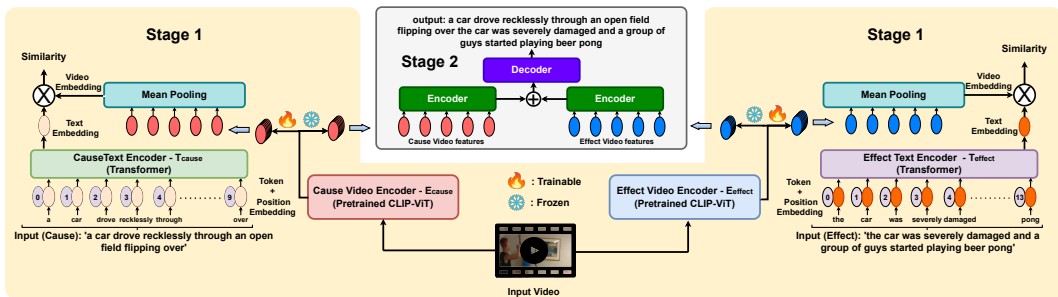

Figure 3: The two-stage Cause-Effect Network (CEN) architecture. Stage 1: Separate Cause ($E_{cause}$) and Effect ($E_{effect}$) video encoders, pretrained using CLIP-ViT, learn specialized video representations. Corresponding text encoders ($T_{cause}$ and $T_{effect}$) encode the cause and effect portions of the CTN caption. Contrastive losses are applied to align the video and text embeddings. Stage 2: The learned cause and effect video features are encoded separately ($Enc_{cause}$ and $Enc_{effect}$) and concatenated before being input to the decoder, which generates the final CTN caption.

sets clear requirements for grounding the captions in the provided descriptive context maintains temporal consistency, and avoids unsupported details. The illustrative examples demonstrate the desired format, facilitating the generation of plainly written, length-constrained Cause and Effect statements that capture the video's causal-temporal narrative. By explicitly outlining instructions and constraints, the carefully designed prompt steers the LLM's generative capabilities towards producing CTN captions. Further details are provided in Appendix A.2 and A.3.

**CTN Caption Generation and Automatic Evaluation:** We send Prompt 1 into the Mixtral of Experts LLM Jiang et al. (2024), which generates CTN captions. The LLM's advanced natural language understanding and generation capabilities, combined with the few-shot prompting approach, enable the production of captions that encapsulate complex causal and temporal relationships. To ensure the quality and relevance of the generated CTN captions, we employ the EMScore Shi et al. (2022) metric for evaluation. EMScore directly measures the consistency between a caption and the video content, and has been shown to be more effective than other metrics de Souza Inácio & Lopes (2023) in evaluating video relevance without referenced captions. We set a threshold ($\theta = 0.2$) for the EMScore value, indicating adequate relevance to the video. Captions with an EMScore above the $\theta$ are kept, while those below the $\theta$ are discarded, and the LLM generates a new caption. This iterative refinement process continues until a caption meets the EMScore threshold, ensuring that the final captions efficiently describe the relevant events in the video. Further details are provided in Appendix A.4.

**Human Evaluation:** To further validate the quality of our CTN captions, we conduct a human evaluation study using standard practice Chen & Dolan (2011); Xu et al. (2016). We randomly sample 100 videos out of 11,970 videos (10,000 MSRVTT-CTN and 1,970 MSVD-CTN) using proportional stratified random sampling, which yields a margin of error of 8.2% at a 90% confidence level. Five independent domain experts evaluate the videos and the CTN captions on three criteria: causal accuracy - the degree to which the caption correctly identifies and describes cause-effect relationships; temporal coherence - the extent to which the caption accurately represents the sequence of events; and relevance - how well the caption reflects the overall content. Each rater does 300 evaluations and each criterion is rated on a 5-point Likert scale. Further details are provided in Appendix A.5.

## 3.2 PROPOSED CAUSE-EFFECT NETWORK (CEN)

Existing SOTA video captioning methods based on LSTMsNadeem et al. (2023), GNNsHendria et al. (2023), and TransformersWang et al. (2022) lack dedicated mechanisms to explicitly model the intricate causal-temporal narrative sequences. As shown in Figure 3, our proposed CEN employs a two-stage approach to address this limitation (see Section 3. In Stage 1 (Section 3.2.1), CEN learns specialized cause and effect video representations using separate encoders trained on the corresponding portions of the CTN captions. The input is raw video frames from the entire video without requiring explicit segmentation of cause-effect regions, and the output is the learned cause and effect features extracted from the encoders. The learned video features from Stage 1 are then passed to Stage 2 (Section 3.2.2), which utilizes a sequence-to-sequence transformer-based network with two new encoders to generate the final captions. The parameters of stage 1 encoders are frozen at this stage. The cause and effect video features are separately encoded and then concatenated before being input to the decoder. This allows the model to synthesize the specialized representations

learned in Stage 1 and generate captions that accurately capture the causal and temporal relationships in the videos.

### 3.2.1 STAGE 1: CAUSAL VISUAL ENCODING

For the CEN network to learn causal-temporal narrative, we design two separate visual encoding models trained on two parts of the captions (see Figure 3).
(1) **Cause Video Encoder** $E_{cause}$: Trained on the cause captions part describing the initiating events;
(2) **Effect Video Encoder** $E_{effect}$: Trained on the effect part of the captions capturing the consequential outcomes.
We instantiate $E_{cause}$ and $E_{effect}$ as two separate instances of the CLIP-ViT (ViT-B/32) model Radford et al. (2021), adapted from CLIP4Clip Luo et al. (2022) (see Appendix A.6 for more details). For both Cause and Effect text encoders ($T_{cause}$ and $T_{effect}$ respectively), we employ a 12-layer transformer with a hidden size of $512$ and $8$ attention heads, using the weights from the pre-trained CLIP Radford et al. (2021) text encoder. As in CLIP Radford et al. (2021), the [EOS] token's representation from the last layer of the text encoders is used as the feature representation for the input text. Similar to CLIP4Clip, we use mean pooling for the video embedding and then, adopt cosine similarity to measure the similarity between the video embedding (mean pooled) and the text embedding. Given a video embedding $\hat{r}_i$ and a text embedding $t_j$, the similarity function is defined as $s(\hat{r}_i, t_j) = \frac{t_j^\top \hat{r}_i}{|t_j||\hat{r}_i|}$. During training, each model is optimized using a contrastive loss, as in CLIP Radford et al. (2021), over its respective cause/effect portion from CTN captions. For a batch of $N$ video-cause text pairs $(v_i, c_i)$ and video-effect text pairs $(v_i, e_i)$, the contrastive losses for the cause ($\mathcal{L}_{cause}$) and effect ($\mathcal{L}_{effect}$) video encoders are defined in 1 and 2 respectively:

$$\mathcal{L}_{cause} = \mathcal{L}_{v2t}(E_{cause}(v_i), T_{cause}(c_i)) + \mathcal{L}_{t2v}(E_{cause}(v_i), T_{cause}(c_i)), \quad (1)$$
$$\mathcal{L}_{effect} = \mathcal{L}_{v2t}(E_{effect}(v_i), T_{effect}(e_i)) + \mathcal{L}_{t2v}(E_{effect}(v_i), T_{effect}(e_i)), \quad (2)$$

where $\mathcal{L}_{v2t}$ and $\mathcal{L}_{t2v}$ are the video-to-text and text-to-video contrastive losses, respectively, defined as:

$$\mathcal{L}_{v2t} = -\frac{1}{N} \sum_{i=1}^{N} \log \frac{\exp(s(\hat{r}_i, t_i))}{\sum_{j=1}^{N} \exp(s(\hat{r}_i, t_j))}, \quad (3) \qquad \mathcal{L}_{t2v} = -\frac{1}{N} \sum_{i=1}^{N} \log \frac{\exp(s(\hat{r}_i, t_i))}{\sum_{j=1}^{N} \exp(s(\hat{r}_j, t_i))} \quad (4)$$

After training, we freeze the weights of $E_{cause}$ and $E_{effect}$ for the second stage, where we extract the cause and effect video features, respectively.

### 3.2.2 STAGE 2: CTN CAPTION GENERATION

In the second stage (see Figure 3), we employ a sequence-to-sequence transformer-based network, consisting of two separate encoders ($Enc_{cause}$ and $Enc_{effect}$) for processing cause and effect video features, respectively, and a decoder $Dec$ for generating the final CTN captions. The encoded cause and effect video representations are concatenated before being sent to the decoder. Both encoders and the decoder are initialized with the weights from the pre-trained Uni-VL model Luo et al. (2020). Given a video $v$, we first extract the cause and effect video features using the pre-trained $E_{cause}$ and $E_{effect}$ encoders:

$$F_{cause} = E_{cause}(v), \quad (5) \qquad\qquad F_{effect} = E_{effect}(v). \quad (6)$$

The extracted features $F_{cause}$ and $F_{effect}$ are separately encoded using $Enc_{cause}$ and $Enc_{effect}$, each consisting of two transformer layers, to obtain the respective encoded representations: $h_{cause} = Enc_{cause}(F_{cause})$ and $h_{effect} = Enc_{effect}(F_{effect})$.
The encoded cause and effect representations are concatenated to form a single representation $h_{concat} = [h_{cause}; h_{effect}]$, which is then input to the decoder $Dec$, consisting of two transformer layers, to generate the CTN caption $\hat{y}$ by attending to $h_{concat}$ at each time step $t$:

$$\hat{y}_t = Dec(h_{concat}], \hat{y}_{<t}), \quad (7)$$

where $\hat{y}_{<t}$ denotes the previously generated words. For training and evaluation, we combine the cause and effect parts of the ground-truth CTN caption with a space in between to form a single, coherent caption. We opt for space separation to maintain consistency with the raw input format of the training datasets (MSRVTT and MSVD), which contain no separators and also, for a fair comparison with the SOTA. This combined caption serves as the target for the decoder during training and the reference for evaluating the generated captions using diffferent evaluation metrics. The entire network is trained end-to-end using the cross-entropy loss between the generated caption $\hat{y}$ and the ground-truth CTN caption $y$:

$$\mathcal{L}_{caption} = -\frac{1}{T} \sum_{t=1}^{T} \sum_{c=1}^{|V|} y_{t,c} \log(\hat{y}_{t,c}), \quad (8)$$

where: $T$ is the length of the caption. $|V|$ is the size of the vocabulary. $y_{t,c}$ is the ground-truth label for the $c$-th word at time step $t$ (1 if the word is present, 0 otherwise). $\hat{y}_{t,c}$ is the predicted probability of the $c$-th word at time step $t$. Finally, our total loss $\mathcal{L}_{total}$ is:

$$\mathcal{L}_{total} = \mathcal{L}_{cause} + \mathcal{L}_{effect} + \mathcal{L}_{caption} \qquad (9)$$

This two-stage approach first encodes the causal aspects into the video representations, facilitating the generation of coherent causal-temporal narrative descriptions in the second stage (see Experiments section 4) to generate CTN captions as final output.

## 4 EXPERIMENTS

We conduct extensive experiments to validate our proposed CEN architecture on the new CTN video captioning benchmark dataset. We present comprehensive evaluation against the state-of-the-art video captioning methods and vision-language models (VLMs) demonstrating the superior performance of the proposed network on the CTN benchmark dataset.

### 4.1 EVALUATION METRICS

For quantitative comparison, we use three metrics as per Celikyilmaz et al. (2020); CIDEr (C) Vedantam et al. (2015); Shi et al. (2022): Measures alignment of generated captions with references by highlighting frequently occurring terms, capturing the ability to reproduce salient causal-temporal narrative elements. ROUGE-L (R-L) Lin (2004): Evaluates the longest common subsequence between generated and reference captions, considering both precision and recall for assessing semantic similarity, including temporal dynamics and causal relationships. SPICE (S) Anderson et al. (2016): Evaluates semantic quality by considering the overlap of scene graphs between generated and reference captions, effectively assessing the ability to capture causal relationships and event sequences.

### 4.2 IMPLEMENTATION DETAILS

We generate the CTN captions using the Mixtral of Experts LLM Jiang et al. (2024), running on A100-80GB GPUs. For the SOTA models, we follow the hyperparameter settings specified in their respective methods for training on the MSR-VTT and MSVD datasets. Our CEN model is trained using the Adam optimizer with learning rates of $1 \times 10^{-4}$ (stage 1) and $1 \times 10^{-6}$ (stage 2) and a batch size of 64 for 10 epochs (stage 1) and 50 epochs (stage 2). For comparison with recent Vision-Language Models (VLMs), we fine-tune VideoLLaVA Lin et al. (2023) and ShareGPT4Video Chen et al. (2024) using both LoRA and simple fine-tuning approaches on our CTN benchmark dataset. We use the recommended hyperparameters for each model during fine-tuning. Further details are provided in the Appendix A.1.

### 4.3 RESULTS AND DISCUSSION

We comprehensively evaluate the performance of our proposed CTN generation pipeline and also, the CEN architecture against several SOTA methods, including SEM-POS Nadeem et al. (2023), AKGNN Hendria et al. (2023), and GIT Wang et al. (2022) and the VLMs VideoLLaVA Lin et al. (2023) and ShareGPT4Video Chen et al. (2024) on our generated CTN captions benchmark datasets.

#### 4.3.1 QUANTITATIVE RESULTS

**CTN Captions Benchmark Dataset:** We generate our CTN captions (1 caption per video) using two widely-used video captioning datasets: 1) MSRVTT Xu et al. (2016), consists of 10,000 videos with 20 human-annotated captions per video, and MSVD Chen & Dolan (2011), with 1,970 videos focused on human activities with approx. 50 captions per video. The length of the caption is on avg. 19 words (cause=10 words, effect=9 words) for MSRVTT-CTN and 17 words (cause=8 words, effect=7 words) for MSVD-CTN. Then, we train as well as test our CEN, SOTA methods and VLMs on the MSVRTT-CTN and MSVD-CTN.

Our CTN generation approach generates good quality captions with 52.1% CTN captions exceeding 0.27 EMScore (threshold=0.2). Then, in the human evaluation study, the overall mean quality score of 4.8 ($\sigma = 0.40$) on a 5-point Likert scale, indicates high performance across all criteria; 93% of captions receive scores of 4 or higher across all three dimensions, with 84% achieving perfect scores. To ensure reliability across raters, we calculate the intraclass correlation coefficient (ICC). The ICC for absolute agreement among raters is 0.87 (95% CI: 0.83-0.91), indicating high inter-rater reliability. These results validate our automatic generation and evaluation process, demonstrating that our CTN captions benchmark dataset provides high-quality, coherent representations of causal-temporal narratives in videos. Further details are provided in Appendix A.5.

Table 1: Comparison of our CEN architecture against SOTA methods and VLMs on the MSVD-CTN and MSRVTT-CTN datasets. The best results in each category are in bold. R-L, C, and S denote ROUGE-L, CIDEr, and SPICE scores, respectively.

| Method | MSVD-CTN | | | MSRVTT-CTN | | |
|---|---|---|---|---|---|---|
| | R-L (↑) | C (↑) | S (↑) | R-L (↑) | C (↑) | S (↑) |
| SEM-POS | 25.39 | 37.16 | 14.46 | 20.11 | 26.01 | 12.09 |
| AKGNN | 25.11 | 35.08 | 14.55 | 21.42 | 25.90 | 11.99 |
| GIT | 27.51 | 45.63 | 15.58 | 24.51 | 32.43 | 13.70 |
| VideoLLaVA (Zero-shot) | 21.80 | 30.55 | 14.67 | 19.33 | 16.24 | 12.49 |
| VideoLLaVA (LoRA FT) | 24.56 | 34.98 | 15.41 | 21.21 | 18.97 | 13.28 |
| VideoLLaVA (Simple FT) | 25.61 | 36.12 | 16.09 | 22.18 | 19.98 | 13.07 |
| ShareGPT4Video (Zero-shot) | 21.66 | 27.06 | 14.06 | 20.27 | 17.08 | 12.21 |
| ShareGPT4Video (LoRA FT) | 24.39 | 30.72 | 14.83 | 22.09 | 19.83 | 13.02 |
| ShareGPT4Video (Simple FT) | 25.32 | 31.67 | 14.92 | 23.01 | 20.76 | 13.28 |
| **CEN (Ours)** | **31.46** | **63.51** | **19.25** | **27.90** | **49.87** | **15.76** |

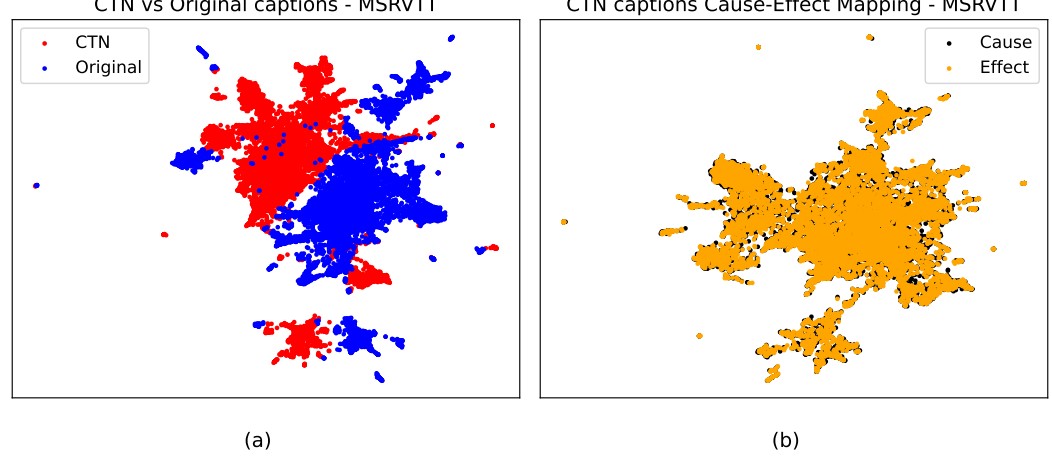

Figure 4: (a) UMAP visualization of video features learned from CTN (red) and original (blue) captions on MSR-VTT, showing non-overlapping feature spaces. (b) UMAP visualization of video features learned from cause (black) and effect (orange) parts of CTN captions on MSRVTT-CTN, showing near-complete overlap.

**Cause-Effect Network (CEN):** CEN outperformed all SOTA methods and the VLMs across all metrics and datasets as shown in Table 1. On MSVD-CTN, CEN surpassed the next best (GIT) by 3.95 ROUGE-L, 17.88 CIDEr, and 3.67 SPICE points. On MSRVTT-CTN, CEN led GIT by 3.39 ROUGE-L, 17.44 CIDEr, and 2.06 SPICE points. These significant gains highlight CEN's effectiveness in capturing causal narratives and temporal dynamics.

We compare CEN's performance against zero-shot and fine-tuned (FT) versions of VideoLLaVA and ShareGPT4Video. CEN consistently outperforms these models, even after fine-tuning, demonstrating the effectiveness of our specialized architecture for causal-temporal narrative generation. For example, on MSRVTT-CTN, CEN achieves a CIDEr score of 49.87, compared to 19.98 for VideoLLaVA (Simple FT) and 20.76 for ShareGPT4Video (Simple FT).

To gain further insights into the effectiveness of the CTN captions in capturing causal-temporal narratives, we visualize the video feature representations learned by CLIP-ViT encoders trained on CTN and Original captions using UMAP McInnes et al. (2018) dimensionality reduction from high dimension onto a 2D plane. In Figure 4(a), we compare the representations learned from CTN captions (red) and Original captions (blue) on the MSR-VTT dataset. The non-overlapping feature spaces indicate that CTN captions capture the causal and temporal relationships not present in the original captions. In Figure 4(b), we visualize the representations learned from the cause (black) and effect (orange) parts of the CTN captions. The near-complete overlap suggests a strong correlation between the cause and effect components, aligning with the inherent structure of causal-temporal narratives and supporting the design of the CEN architecture.

Table 2: Ablation study results on the MSVD-CTN and MSRVTT-CTN datasets. $E_{combined}$, w/o FT CLIPs, Only $E_{cause}$ and Only $E_{effect}$ are baselines of our CEN architecture, while Zero Shot X and Fine-tune X represent cross-dataset evaluation settings. The best results in each category are in bold.

| Method | MSVD-CTN | | | MSRVTT-CTN | | |
|---|---|---|---|---|---|---|
| | R-L ($\uparrow$) | C ($\uparrow$) | S ($\uparrow$) | R-L ($\uparrow$) | C ($\uparrow$) | S ($\uparrow$) |
| $E_{combined}$ | 30.93 | 55.72 | 17.04 | 27.34 | 45.97 | 15.07 |
| w/o FT - single CLIP | 27.81 | 46.23 | 15.82 | 25.34 | 32.31 | 14.27 |
| w/o FT - Two CLIP | 28.40 | 53.84 | 16.58 | 26.10 | 40.92 | 14.55 |
| Only $E_{cause}$ | 30.72 | 56.42 | 18.43 | 27.19 | 47.10 | 15.21 |
| Only $E_{effect}$ | 30.70 | 57.14 | 17.89 | 27.24 | 45.58 | 15.19 |
| Zero Shot X | 27.16 | 39.65 | 14.45 | 24.73 | 29.76 | 12.26 |
| Fine-tune X | **31.78** | **65.60** | **19.39** | 27.47 | 47.74 | 15.65 |
| CEN (Ours) | 31.46 | 63.51 | 19.25 | **27.90** | **49.87** | **15.76** |

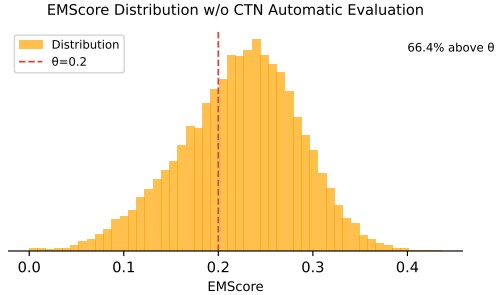

Figure 5: EMScore distribution of initial CTN captions showing 66.4% exceed quality threshold $\theta$=0.2

### 4.3.2 ABLATION RESULTS

**CTN Captions Benchmark Dataset:** Figure 5 demonstrates the distribution of caption quality without automatic evaluation filtering. The EMScore distribution across 11,970 videos shows 66.4% of initially generated captions achieve scores above $\theta$=0.2, with a peak around EMScore=0.24. The remaining 33.6% of captions fall below our quality threshold, highlighting the importance of our automatic evaluation and regeneration process in maintaining CTN quality.

**Cause-Effect Network (CEN):** To examine the effectiveness of CEN, we compare it with 6 baselines in Table 2:

**a) $E_{combined}$:** Instead of using two separate video encoders for cause and effect, we train one CLIP-ViT using the combined cause and effect captions separated by space for the text encoder. The performance drops across all metrics and datasets compared to the CEN architecture, which underscores the benefits of dedicated encoders for capturing cause and effect dynamics.

**b) w/o FT - Single CLIP:** This ablation is performed using one CLIP encoder and no Fine-tuning (FT) on cause, effect, or cause+effect (combined). The results show the effectiveness of CTN fine-tuning.

**c) w/o FT - Two CLIP:** This ablation is performed using two CLIP $E_{cause}$ and $E_{effect}$ encoders (as in our CEN) without performing the Stage 1 fine-tuning (FT) on cause and effect. This demonstrates the effectiveness of CTN fine-tuning and also, the performance increase in comparison to "w/o FT - Single CLIP" demonstrates the effectiveness of CEN.

**d) Only $E_{cause}$ and Only $E_{effect}$:** These ablations employ only the cause encoder or only the effect encoder, respectively. While retaining reasonable performance, both variants fall short of the performance of the full model across all metrics and datasets.

**e) Zero Shot X:** In this zero-shot setting, we evaluate the models trained on one dataset (MSVD-CTN or MSRVTT-CTN) against the other dataset, without any fine-tuning on the other dataset. The results are either better than or comparable with SOTA and VLMs in Table 1.

**f) Fine-tune X:** Similar to the Zero Shot X, this variant involves fine-tuning the models trained on cross-datasets. Notably, this fine-tuning process leads to improved performance on the MSVD dataset. This observation demonstrates the potential for transfer learning of CEN model from larger datasets i.e. MSRVTT-CTN and the ability to leverage their knowledge effectively on smaller datasets i.e. MSVD-CTN through fine-tuning Lin et al. (2022); Ventura et al. (2024)

Overall, the ablation study validates the efficacy of our proposed CEN architecture and approach, emphasizing the significance of dedicated encoders for capturing cause and effect relationships independently as the performance drops are observed in $E_{combined}$, w/o FT - Single CLIP, w/o FT - Two CLIP, Only $E_{cause}$, and Only $E_{effect}$ baselines. We also provide LLM-based evaluations in Appendix A.9.

### 4.3.3 QUALITATIVE RESULTS

Qualitative examples, in Figure 6, demonstrate CEN's strength in accurately articulating causal relationships and event sequences, while SOTA methods and recent VLMs struggle. For example:
**(a) Video game scene:** CEN accurately captures the causality ("performs a fatality move" causing "killing another"), while other methods miss this crucial relationship. SEM-POS and GIT provide overly simplistic descriptions, while AKGNN generates irrelevant details. VLMs (Video-LLaVA and ShareGPT4Video) misinterpret the scene entirely.; **(b) Paper folding:** CEN correctly identifies both the cause (folding paper) and effect (creating a paper airplane). Other methods either miss

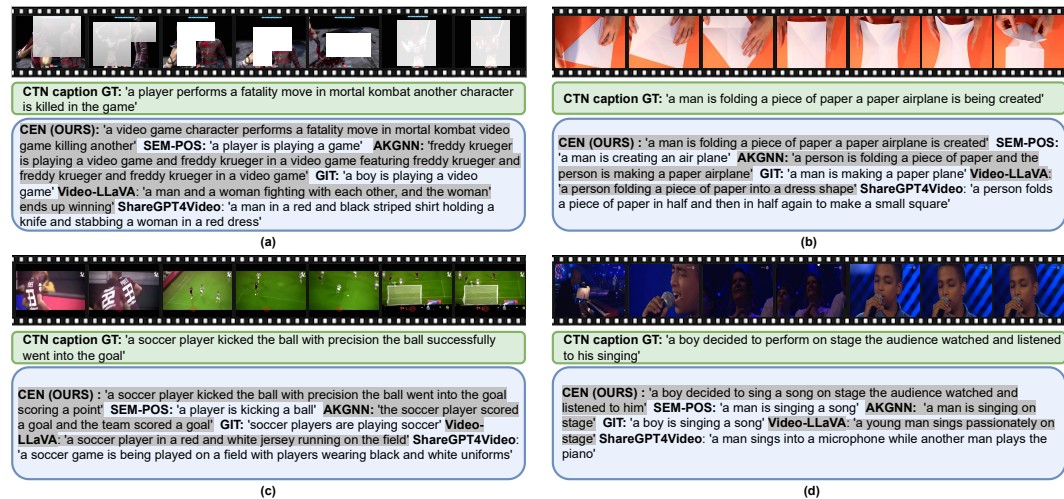

Figure 6: Qualitative examples across scenarios like video games, paper folding, soccer, and singing. CEN (Ours) captions accurately capture causal narratives and temporal sequences from ground truth, outperforming SOTA video captioning methods.

the causal relationship or misinterpret the action (e.g., ShareGPT4Video's "small square" instead of an airplane).; **(c) Soccer scene:** CEN accurately links the precise kick to the goal, capturing both cause and effect. Other methods either focus on a single action or provide generic descriptions of a soccer game.; **(d) Singing performance:** CEN captures the boy's decision to perform (cause) and the audience's reaction (effect). Other methods mostly describe the act of singing without the causal-temporal context.

Overall, quantitative and qualitative results showcase CEN's superior performance in understanding and generating causal-temporal narrative video captions compared to existing SOTA methods and recent VLMs. More results are provided in Appendix A.10.

## 5 LIMITATIONS

While our work represents a significant step forward in causal-temporal narrative video captioning, complex onvoluted causal relationships are still a challenge. This necessitates further architectural enhancements for improved generalization. Explicit integration of spatial reasoning, multi-agent interactions, and long-term dependencies could further enhance the robustness and applicability of our approach. Despite these limitations, our work opens up new avenues for innovative applications and research directions, further solidifying the importance of causal-temporal narrative understanding in video analysis tasks.

## 6 CONCLUSION

In this paper, we introduced for the first time a Causal-Temporal Narrative (CTN) captions benchmark dataset and proposed a novel Cause-Effect Network (CEN) tailored for causal-temporal narrative video captioning. This work finds vast applications in automated video summarization, question-answering, assistive technologies, surveillance, and educational content creation. The CTN captions benchmark dataset provides a comprehensive testbed for evaluating video understanding models' ability to grasp complex temporal and causal dynamics, which will be released for research purpose on https://narrativebridge.github.io/. And CEN explicitly models cause-effect relationships and temporal dynamics to generate rich, contextually relevant descriptions capturing nuanced causal-temporal narrative in videos, demonstrating significant performance improvement over SOTA methods (Section 4). NarrativeBridge lays the foundation for a paradigm shift, where models comprehend underlying causal-temporal narrative driving events, unlocking new frontiers in contextually aware human-machine interactions with video.

For future work, we aim to integrate CTN caption generation with existing image captioning techniques, to annotate unlabeled videos with causal-temporal narrative labels. A few frames of the video will be labelled using off-the-shelf image captioning methods, and CTN caption generation will exploit the labelled frames to generate one coherent caption for the unlabelled video (see Appendix A.8). This synergistic approach opens new avenues for comprehensive video understanding and annotation, enabling more robust and accurate video analysis pipelines.

## ACKNOWLEDGEMENT

This research was partly supported by the British Broadcasting Corporation Research and Development (BBC R&D), Engineering and Physical Sciences Research Council (EPSRC) Grant EP/V038087/1 "BBC Prosperity Partnership: Future Personalised Object-Based Media Experiences Delivered at Scale Anywhere".

## ETHICS STATEMENT

Our work on NarrativeBridge focuses on improving video captioning through causal-temporal narrative understanding. We use publicly available datasets (MSR-VTT and MSVD) and have cited their creators appropriately. We will release our new CTN captions benchmark dataset on our project webpage https://narrativebridge.github.io/, where we also mention the licenses of all assets used. Our research does not involve human subjects or crowdsourcing, and we do not use or curate data containing personally identifiable information or offensive content. We have read and adhered to the ethics review guidelines for ICLR submissions. We acknowledge potential ethical concerns (e.g., privacy in surveillance, risk of misleading content) but are committed to responsible development.

## REPRODUCIBILITY STATEMENT

We will release the CTN captions benchmark dataset and CEN model weights upon paper publication on our project website https://narrativebridge.github.io/. We provide comprehensive information on datasets (Section 4.3.1), evaluation metrics (Section 4.1), and implementation details (Section 4.2 and Appendix A.1). We specify data splits, hyperparameters, and other key details necessary for reproducing our results. We include information about the type of GPUs and compute resources used in Section 4.2 and Appendix A.1. We properly cite all existing datasets (MSR-VTT, MSVD) and models (CLIP-ViT, Uni-VL) used in our work.

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

# A   APPENDIX

## A.1   IMPLEMENTATION DETAILS

We first extract frames from the video clip and then encode them using the Cause and Effect Video Encoders. We adopt a sampling strategy where we sample one frame per second, with a maximum of 20 frames. This approach ensures that the sampled frames cover the entire duration of the video, providing a comprehensive representation of the video content while maintaining computational efficiency. For the fine-tuning experiment in Table 2, all the hyperparameters remain the same except the learning rate i.e. 0.0000005. For our experimentation of the Cause and Effect Network (CEN), we use standard splits of the MSVD Chen & Dolan (2011) and MSRVTT Xu et al. (2016) datasets. For MSVD, we use 1200 videos for training, 100 for validation, and 670 for testing. For MSRVTT, we use 6513 videos for training, 497 for validation, and 2990 for testing. These splits are commonly used in video captioning research to ensure fair comparison with other methods. All the experiments in Stage 1 and Stage 2 of our CEN are run using A100-80GB and RTX 3090-24GB GPUs respectively. We implement SEM-POS Nadeem et al. (2023), AKGNN Hendria et al. (2023) and GIT Wang et al. (2022) using RTX 3090-24GB, A100-80GB and A100-80GB GPUs respectively.
For comparison with Vision-Language Models (VLMs), we implement two fine-tuning approaches: LoRA Fine-Tuning and Simple Fine-Tuning using A100-80GB GPUs. LoRA Fine-Tuning is applied specifically to the LLM component, with a learning rate of 2e-4 for LoRA parameters. Simple Fine-Tuning is applied to the entire model, using an AdamW Loshchilov (2017) optimizer with a cosine learning rate schedule (initial learning rate: 1e-3, warmup ratio: 0.03).

## A.2   PROMPT DESIGN PROCESS

In Section 3.1, we introduce our approach for generating Causal-Temporal Narrative (CTN) captions using a large language model (LLM) and few-shot prompting. The prompt design played a crucial role in guiding the LLM to generate captions that accurately capture the cause-effect relationships and temporal dynamics in the video content. Figure 1 showcases the effectiveness of our few-shot based prompt, which resulted in a coherent and contextually relevant CTN caption for the given video. To further illustrate the importance of prompt design and the benefits of few-shot learning, we conduct additional experiments with zero-shot prompting. In these experiments, we evaluate the performance of the LLM in generating CTN captions without providing any example captions in the prompt.
**Zero-Shot Prompting**: Zero-shot prompting refers to the process of providing large language models (LLMs) with a task prompt without any accompanying examples or demonstrations, requiring the model to generate relevant responses based solely on its pre-trained knowledge and understanding of the prompt. First, we aim to evaluate the performance of LLMs in generating causal-temporal narrative (CTN) captions for videos without providing any few-shot examples.
To achieve this, we construct 4 distinct prompts, each with varying degrees of quality, and experimented with all of them to assess their effectiveness in a zero-shot setting, starting with the most challenging prompt and progressively improving the quality. Prompt 2 presents an extremely terse and poorly structured set of instructions. It contains minimal context and information, merely instructing the model to "Make Cause Effect thing from captions" and listing a set of numbered requirements without providing clear explanations. As expected, the resulting captions (Figure 7) are of low quality and often lack coherence and relevance to the video content. Moreover, the LLM generated multiple captions instead of a single coherent caption, and the output format does not adhere to the specified structure (a single JSON object containing the Cause and Effect statements).

```
Make Cause Effect thing from captions. Follow: 1. Use only caption info 2. Be relevant 3. No new stuff
4. No conclusions 5. Same time 6. Plain English short sentences 7. 15 words max 8. Just JSON.
[Descriptive Captions]:
<descriptive_captions>
[Causal Temporal Narrative]:
```

Prompt 2: Zero-shot LLM Prompt with minimal context and unexplained requirements for generating cause-effect from captions.

```
  { "1":  {"Cause":  "A car crashes", "Effect":  "Guys play beer pong"},
 "2":  {"Cause":  "A car drives through an open field", "Effect":  "Kicks up
 dirt"},
 "3":  {"Cause":  "Unclear", "Effect":  "A car flips over"},
 "4":  {"Cause":  "Unclear", "Effect":  "A car gets wracked"},
 "5":  {"Cause":  "Unclear", "Effect":  "A car is being flipped over"},
 "6":  {"Cause":  "A dirt vehicle rides", "Effect":  "It rolls"},
 "7":  {"Cause":  "Unclear", "Effect":  "A dune buggy flips over"},
 "8":  {"Cause":  "Unclear", "Effect":  "A four wheeler wrecks"},
 "9":  {"Cause":  "A monster truck flips on its side", "Effect":  "Several young
 men shout while playing beer pong"},
 "10":  {"Cause":  "A person drives an offroad car around a field", "Effect":
 "Unclear"},
 "11":  {"Cause":  "A person flips a go kart", "Effect":  "A crowd cheers"},
 "12":  {"Cause":  "Unclear", "Effect":  "A race truck is crashing"},
 "13":  {"Cause":  "A truck rolls over itself", "Effect":  "Boys cheer on a
 friend"},
 "14":  {"Cause":  "Unclear", "Effect":  "A truck tumbles over on itself"},
 "15":  {"Cause":  "A tumbler crashes on a dirt road", "Effect":  "Then a group of
 guys play beer pong"},
 "16":  {"Cause":  "A type of monster truck crashes", "Effect":  "Men are shown
 celebrating"},
 "17":  {"Cause":  "Unclear", "Effect":  "A vehicle flips over"},
 "18":  {"Cause":  "Unclear", "Effect":  "An off road vehicle crashes"},
 "19":  {"Cause":  "Unclear", "Effect":  "A car crashes while driving"},
 "20":  {"Cause":  "Footage from a monster truck style event", "Effect":
 "Followed by a frat party"} }
```

Figure 7: Caption generated using LLM for the example in Figure 1 using Prompt 2

We then progressively improve the prompt quality, moving to a more concise set of instructions in Prompt 3 and a relatively well-structured prompt in Prompt 4. While the captions generated by these improved prompts (Figures 8 and 9) show some enhancements, they still fall short in terms of capturing the causal-temporal narrative accurately and consistently. Additionally, the LLM continues to generate multiple captions and deviate from the desired output format.

```
You need to create a concise Cause and Effect scenario in JSON format based on some descriptive
captions about a video. The scenario must follow these rules: stick to the given information, be
relevant, don't add new events/details or draw conclusions, maintain time consistency, use
simple language in short sentences, and limit each statement to 15 words. Don't put any extra text
outside the JSON.
[Descriptive Captions]:
<descriptive_captions>
[Causal Temporal Narrative]:
```

Prompt 3: Zero-shot LLM Prompt with reasonable structure conveying core task of creating cause-effect scenario from captions.

```
  {{"Event":  "Car crashes", "Effect":  "Guys play beer pong"},
 {"Event":  "Car drives in open field", "Effect":  "Dirt is kicked up"},
 {"Event":  "Car flips over", "Effect":  "None specified"},
 {"Event":  "Car gets wracked", "Effect":  "None specified"},
 {"Event":  "Car is flipped over", "Effect":  "None specified"},
 {"Event":  "Dirt vehicle rides and rolls", "Effect":  "None specified"},
 {"Event":  "Dune buggy flips over", "Effect":  "None specified"},
 {"Event":  "Four wheeler wrecks", "Effect":  "None specified"},
 {"Event":  "Monster truck flips", "Effect":  "Young men shout and play beer
 pong"},
 {"Event":  "Person drives offroad car", "Effect":  "None specified"},
 {"Event":  "Person flips go kart", "Effect":  "Crowd cheers"},
 {"Event":  "Race truck crashes", "Effect":  "None specified"},
 {"Event":  "Truck rolls over", "Effect":  "Boys cheer on a friend"},
 {"Event":  "Tumbler crashes on dirt road", "Effect":  "Guys play beer pong"},
 {"Event":  "Monster truck crashes", "Effect":  "Men are shown celebrating"},
 {"Event":  "Vehicle flips over", "Effect":  "None specified"},
 {"Event":  "Off road vehicle crashes", "Effect":  "None specified"},
 {"Event":  "Car crashes while driving", "Effect":  "None specified"},
 {"Event":  "Monster truck event footage", "Effect":  "Frat party is shown"}}
```

Figure 8: Caption generated using LLM for the example in Figure 1 using Prompt 3

```
You are tasked with generating a causal narrative caption for a video based on a series of provided
descriptive captions. Your generated Cause and Effect scenario should be grounded in the given
information, relevant, avoid introducing new details or implying conclusions, maintain temporal
consistency, use plain English indirect sentences, and limit each Cause and Effect statement to
15 words maximum. Do not include any additional text outside the JSON object.
[Descriptive Captions]:
<descriptive_captions>
[Causal Temporal Narrative]:
```

Prompt 4: Zero-shot LLM Prompt clearly describing caption-based cause-effect generation task with specified requirements.

```
  {{"Event":  "Car crashes", "Effect":  "Guys play beer pong"},
{"Event":  "Car drives in open field", "Effect":  "Dirt is kicked up"},
{"Event":  "Car flips over", "Effect":  "None specified"},
{"Event":  "Car gets wracked", "Effect":  "None specified"},
{"Event":  "Car is flipped over", "Effect":  "None specified"},
{"Event":  "Dirt vehicle rides and rolls", "Effect":  "None specified"},
{"Event":  "Dune buggy flips over", "Effect":  "None specified"},
{"Event":  "Four wheeler wrecks", "Effect":  "None specified"},
{"Event":  "Monster truck flips", "Effect":  "Young men shout and play beer
pong"},
{"Event":  "Person drives offroad car", "Effect":  "None specified"},
{"Event":  "Person flips go kart", "Effect":  "Crowd cheers"},
{"Event":  "Race truck crashes", "Effect":  "None specified"},
{"Event":  "Truck rolls over", "Effect":  "Boys cheer on a friend"},
{"Event":  "Tumbler crashes on dirt road", "Effect":  "Guys play beer pong"},
{"Event":  "Monster truck crashes", "Effect":  "Men are shown celebrating"},
{"Event":  "Vehicle flips over", "Effect":  "None specified"},
{"Event":  "Off road vehicle crashes", "Effect":  "None specified"},
{"Event":  "Car crashes while driving", "Effect":  "None specified"},
{"Event":  "Monster truck event footage", "Effect":  "Frat party is shown"}}
```

Figure 9: Caption generated using LLM for the example in Figure 1 using Prompt 4

Finally, we experiment with a zero-shot prompt (Prompt 5) that closely resembles our few-shot based Prompt 1 from Section 3.1, but without the example captions. The captions generated by this prompt (Figure 10) demonstrate improved coherence and relevance compared to the previous zero-shot prompts. However, they still lack the level of detail, accuracy, and contextual understanding exhibited by the captions generated using our few-shot based prompt (Prompt 1). Furthermore, the LLM persists in generating multiple captions.

```
You are an advanced language model tasked with generating causal temporal narrative captions for a
video. However, you cannot directly access the video series of itself. Instead, you will be provided
with a captions that outline the key events and scenes in the video. Your task is to generate a
concise Cause and Effect scenario, based on the information provided in the descriptive captions.
Be careful, your generated Cause and Effect statements should fulfill the following requirements:
1. Your narrative should be grounded in the information provided by the descriptive captions.
2. Cause and Effect scenario is relevant.
3. It should not introduce any new events or details not mentioned.
4. Avoid implying conclusions.
5. Maintain temporal consistency with the provided captions.
6. Use plain English and direct sentences.
7. Cause and Effect statements each limited to a maximum of 15 words.
8. Do not include any additional text before or after the JSON object.
[Descriptive Captions]:
<descriptive_captions>
[Causal Temporal Narrative]:
```

Prompt 5: Zero-Shot Prompting for Causal Narrative Caption Generation

```
{ "Cause and Effect Statements":  [ "Car crashes in field, causing it to flip
over.", "Dune buggy flips, resulting in a wreck.", "Go kart flipped by person,
leading to crowd cheers.", "Race truck crashes, boys cheer in response.", "Truck
tumbles, friends shout while playing beer pong.", "Monster truck crashes, men
celebrate afterwards.", "Off-road vehicle crashes, causing a tumble.", "Car crash
occurs, followed by a game of beer pong." ] }
```

Figure 10: Caption generated using LLM for the example in Figure 1 using Prompt 5

**Prompt 1 Ablations**: Here, we present systematic ablation experiments analyzing the impact of individual requirements in the design of Prompt 1.

```
[Same as Prompt 1 but without rule 1 about using only information from the captions]
```

Prompt 6: Removing the grounding requirement from Prompt 1

```
{ "Cause":  "a reckless driver speeds through a field in an unstable off-road
vehicle", "Effect":  "the vehicle crashes dramatically and a group celebrates
with beer pong nearby" }
```

Figure 11: Caption generated without grounding requirement

Removing the grounding requirement results in the introduction of speculative details that were not present in the source captions. This demonstrates how the grounding requirement maintains factual accuracy.

```
[Same as Prompt 1 but without rule 5 about temporal consistency]
```

Prompt 7: Removing temporal consistency requirement from Prompt 1

```
{ "Cause":  "people celebrate enthusiastically at a gathering", "Effect":  "a
car flips over while driving through an open field" }
```

Figure 12: Caption generated without temporal consistency requirement

The absence of the temporal consistency requirement leads to incorrect event ordering, with the celebration incorrectly positioned before the car crash. This highlights the importance of maintaining proper temporal sequence in causal narratives.

```
[Same as Prompt 1 but without rule 7 about word limit]
```

Prompt 8: Removing word limit requirement from Prompt 1

```
{ "Cause":  "an adventurous individual decided to drive their vehicle recklessly
through an open field without proper consideration for safety or stability,
leading to a dangerous situation", "Effect":  "the vehicle inevitably lost
control and dramatically flipped over, resulting in a serious accident while
nearby, a group of people were engaging in a casual game of beer pong, seemingly
unaware of the incident" }
```

Figure 13: Caption generated without word limit requirement

Removing the word limit results in verbose descriptions with unnecessary elaboration and speculation about circumstances and motivations.

```
[Same as Prompt 1 but without rule 6 about plain English]
```

Prompt 9: Removing plain English requirement from Prompt 1

```
  { "Cause":  "the automobile's operator executed an ill-advised traversal of the
  pastoral terrain", "Effect":  "culminating in vehicular inversion whilst revelry
  ensued in proximity" }
```

Figure 14: Caption generated without plain English requirement

Without the plain English requirement, the model employs unnecessarily complex vocabulary and formal language that reduces clarity and accessibility.

```
[Same as Prompt 1 but without rule 4 about avoiding conclusions]
```

Prompt 10: Removing the requirement to avoid conclusions from Prompt 1

```
  { "Cause":  "driver's obvious inexperience and poor judgment led to dangerous
  driving in field", "Effect":  "predictable crash occurred while others carelessly
  continued their party, showing clear negligence" }
```

Figure 15: Caption generated without avoiding conclusions requirement

Removing the requirement to avoid conclusions leads to subjective interpretations about driver inexperience and partygoer negligence that cannot be definitively determined from the video content.

```
[Same as Prompt 1 but without rule 2 about relevance]
```

Prompt 11: Removing relevance requirement from Prompt 1

```
  { "Cause":  "a vehicle drives through a field on a sunny day and crashes",
  "Effect":  "people enjoy various outdoor activities including beer pong and
  socializing" }
```

Figure 16: Caption generated without relevance requirement

The absence of the relevance requirement results in the inclusion of peripheral details.
This ablation analysis validates the prompt design choices and underscores the importance of maintaining all requirements for generating effective causal-temporal narrative captions. In conclusion, these findings underscore the importance of our few-shot based prompt design in the CTN caption generation process and demonstrate its superiority over zero-shot prompting approaches. By leveraging the power of few-shot learning, we enable the LLM to generate high-quality CTN captions that effectively capture the cause-effect relationships and temporal dynamics in video content, as evidenced by the results presented in Figure 1. Additionally, our few-shot based prompt along with requirements ensures that the generated captions adhere to the specified JSON format, facilitating seamless integration and usability in downstream applications.

## A.3 COMPARISON OF LARGE LANGUAGE MODELS FOR CTN CAPTION GENERATION

To evaluate the effectiveness of large language models (LLMs) in generating high-quality Causal-Temporal Narrative (CTN) captions, we compare the performance of two open source state-of-the-art LLMs at the time of experimentation: Mixtral of Experts Jiang et al. (2024), which we utilized in our CTN caption generation pipeline, and Llama2-70b Touvron et al. (2023).
We provide the original video captions as input to both models and assess the quality of their generated CTN captions. In the video game example (Figure 17), Mixtral of Experts accurately captures the causal relationship between the fatality move performed by one character and the consequent death of another character in the game. In contrast, Llama2-70b focuses on a specific effect (blood spurting from the character's neck) without explicitly linking it to the cause.
For the paper folding tutorial (Figure 18), both models correctly identify the causal relationship between folding the paper and creating a paper craft. However, Mixtral of Experts provides a more precise description, specifying that the paper craft being created is a paper airplane.
In the soccer highlight scenario (Figure 19), Mixtral of Experts successfully captures the causal link between the soccer player's precise kick and the ball successfully going into the goal. Llama2-70b,

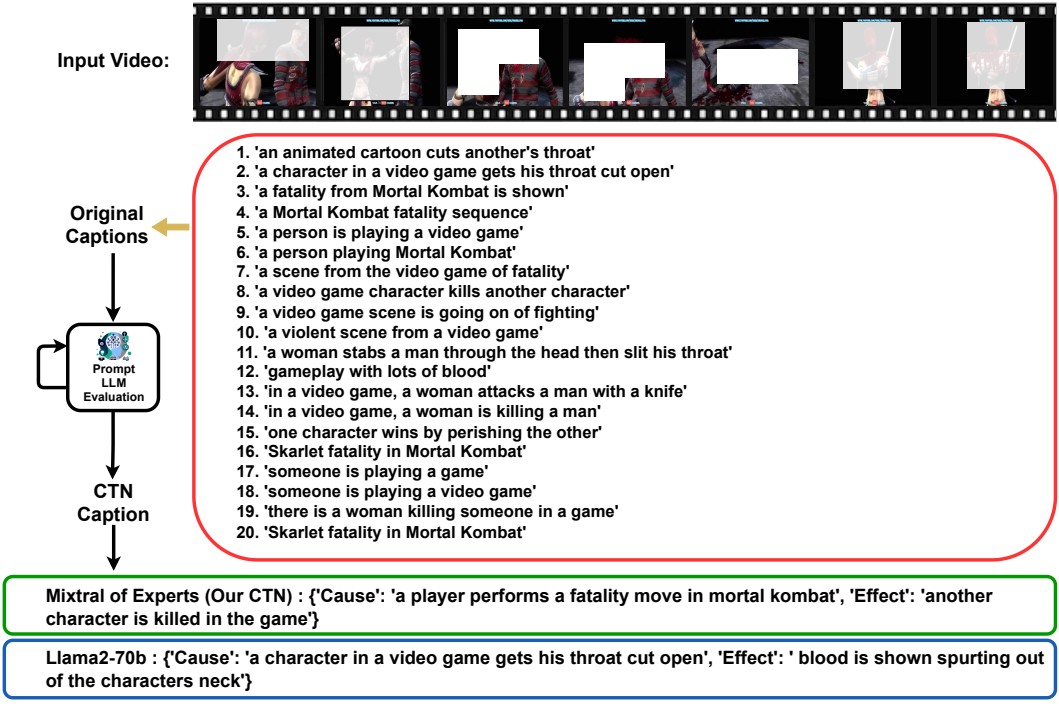

Figure 17: Comparison of Mixtral of Experts and Llama2-70b on a video game sequence involving a fatality move.

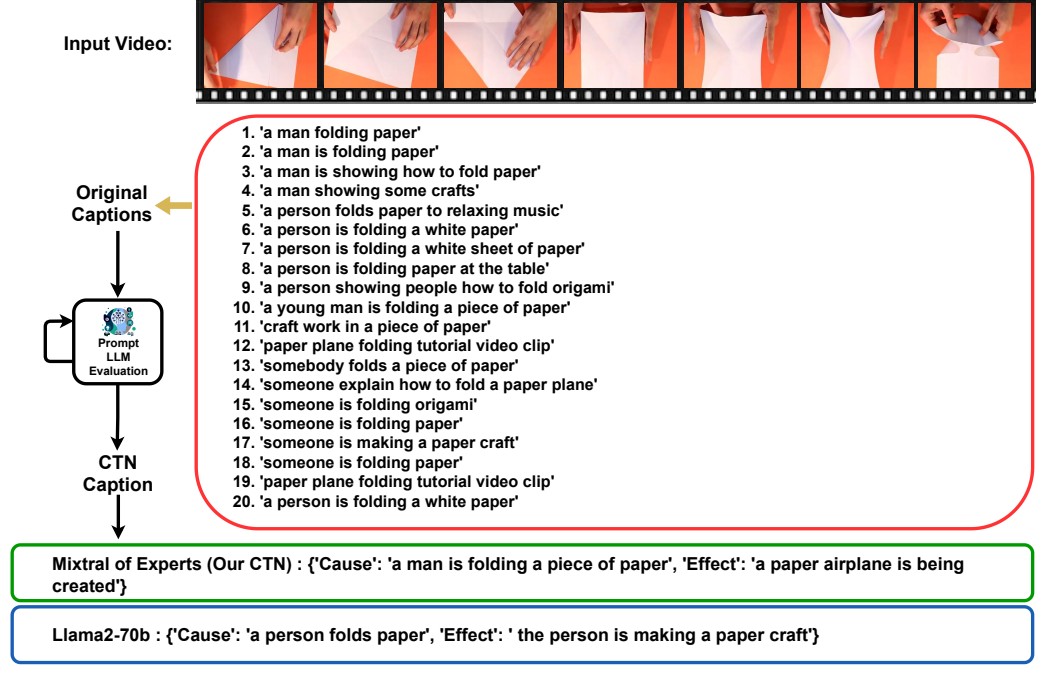

Figure 18: Comparison of Mixtral of Experts and Llama2-70b on a paper folding tutorial to create a paper airplane.

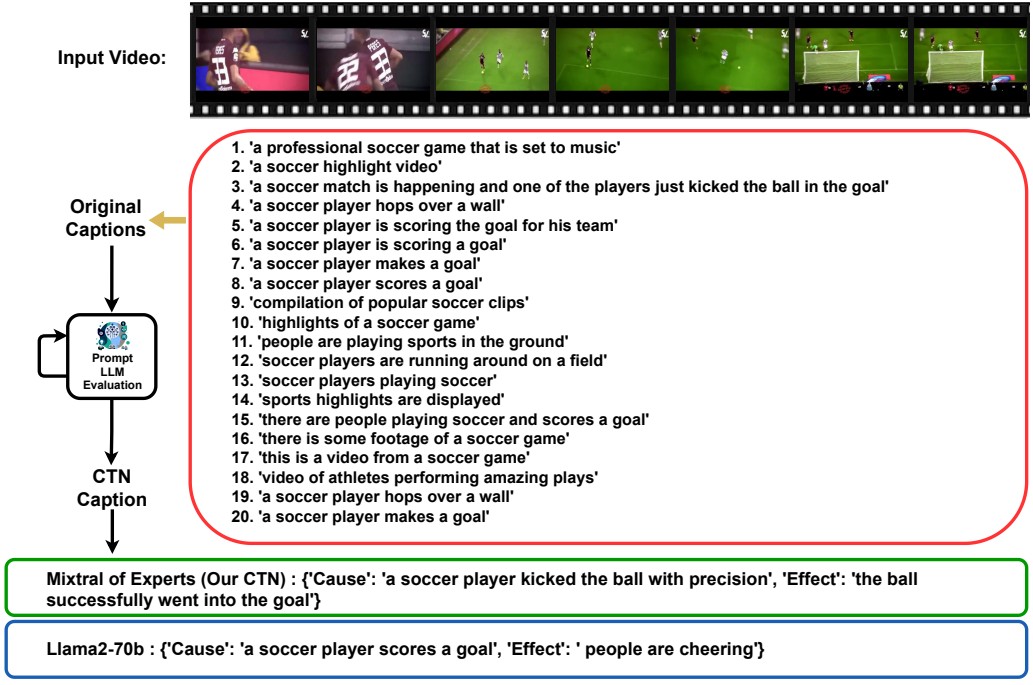

Figure 19: Comparison of Mixtral of Experts and Llama2-70b on highlights from a soccer match.

on the other hand, mentions the effect (people cheering) without explicitly connecting it to the cause (the player scoring a goal).

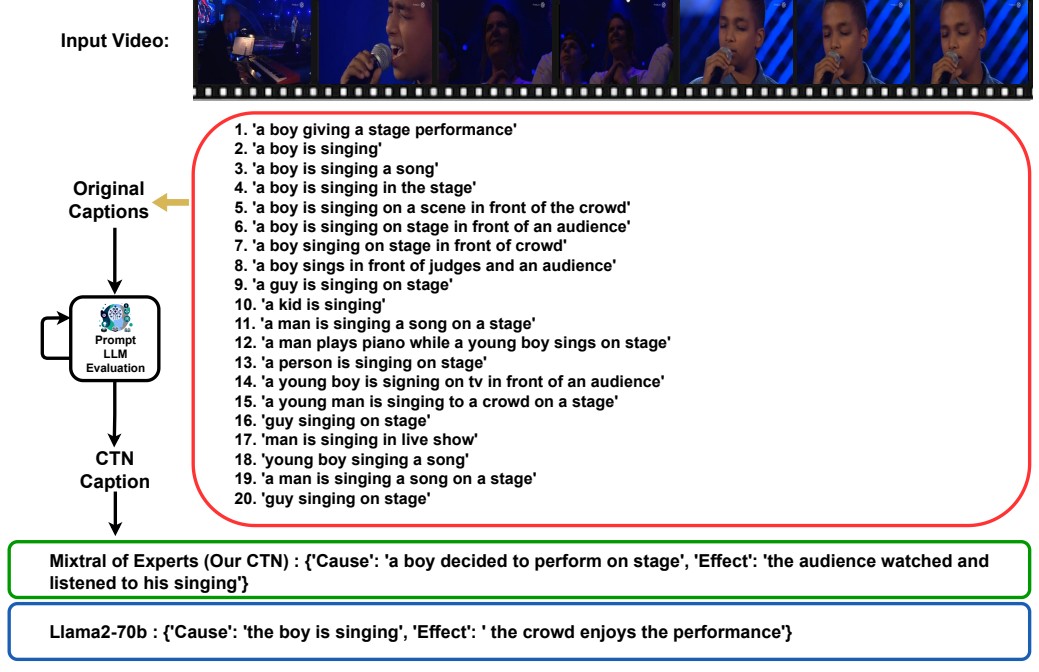

Figure 20: Comparison of Mixtral of Experts and Llama2-70b on a boy singing on stage in front of an audience.

Lastly, for the singing performance (Figure 20), both models accurately describe the cause-effect relationship between the boy's decision to perform on stage and the audience's reaction of watching and listening. However, Mixtral of Experts provides a more detailed description of the audience's response.

Overall, the CTN captions generated by Mixtral of Experts consistently demonstrate a better understanding of the causal-temporal narrative in the videos compared to Llama2-70b. Our approach using Mixtral of Experts effectively captures the cause-effect relationships and temporal dynamics, resulting in more accurate and contextually relevant captions.

These results highlight the importance of selecting an appropriate LLM and designing effective prompts for generating high-quality CTN captions. The superior performance of Mixtral of Experts can be attributed to its architecture and training, which enable it to better understand and articulate the complex causal-temporal narratives present in video content.

### A.4 IMPACT OF AUTOMATIC EVALUATION ON CTN CAPTION GENERATION

In Section 3.1, we discuss the importance of using automatic evaluation to ensure the quality and relevance of the generated Causal Temporal Narrative (CTN) captions. We employ the EMScore metric to measure the consistency between the generated captions and the video content, discarding captions that fell below a predefined threshold. This appendix visually demonstrates the impact of the automatic evaluation step on the quality of the generated CTN captions.

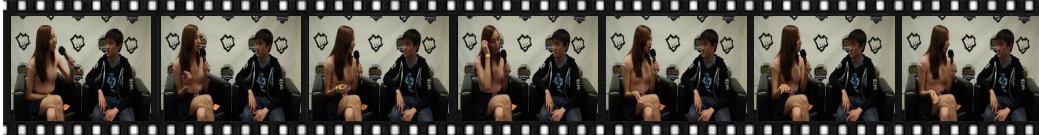

**CTN caption GT - Ours:** {'Cause': 'a female sportscaster interviewed a male athlete', 'Effect': 'she asked him questions'}

**CTN caption (w/o automatic evaluation):** {'Cause': 'a man carelessly neglected taking his prescribed allergy medication', 'Effect': 'he suffered a severe sneezing fit'}

Figure 21: CTN caption comparison for a video of a female sportscaster interviewing a male athlete.

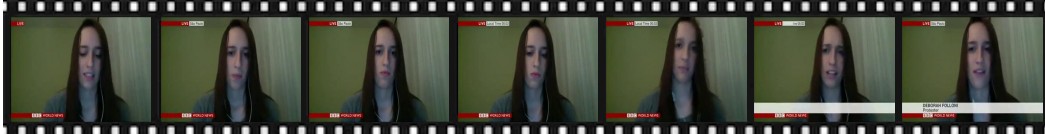

**CTN caption GT - Ours:** {'Cause': 'bus ticket prices increased in Brazil', 'Effect': 'a woman was interviewed about her participation in the protests'}

**CTN caption (w/o automatic evaluation):** {'Cause': 'protest caused by a rise in bus tickets', 'Effect': 'mass gathering of people in rio de janeiro and sao paulo'}

Figure 22: CTN caption comparison for a video of protests in Brazil due to increased bus ticket prices.

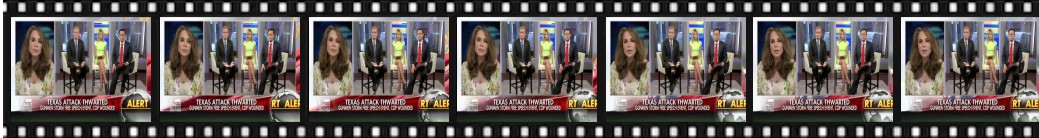

**CTN caption GT - Ours:** {"Cause": "a terror attack occurred in Texas", "Effect": "news anchors discussed freedom of speech issues"}

**CTN caption (w/o automatic evaluation):** {'Cause': 'a band performs a song on stage', 'Effect': 'the audience cheers and enjoys the music'}

Figure 23: CTN caption comparison for a video discussing a terror attack in Texas.

Figure 21 shows an example where the CTN caption generated with automatic evaluation accurately captures the causal relationship between the female sportscaster interviewing the male athlete and her asking him questions. In contrast, the caption generated without automatic evaluation is irrelevant to the video content, discussing a man neglecting his allergy medication and suffering a sneezing fit.

**CTN caption GT - Ours:** {"Cause": "the Air Force used a surveillance system in Iraq", "Effect": "a woman talked about the man who created it"}

**CTN caption (w/o automatic evaluation):** {'Cause': 'a group starts dancing at an event', 'Effect': 'the atmosphere becomes more lively'}

Figure 24: CTN caption comparison for a video about the Air Force using a surveillance system in Iraq.

Similarly, in Figure 22, the CTN caption generated with automatic evaluation correctly identifies the cause of the protests in Brazil as the increase in bus ticket prices and links it to the effect of a woman being interviewed about her participation in the protests. The caption generated without automatic evaluation, while mentioning the protest and its cause, fails to capture the specific effect of the woman's interview. Figure 23 demonstrates how the CTN caption generated with automatic evaluation accurately describes the cause of a terror attack in Texas and its effect on news anchors discussing freedom of speech issues. The caption generated without automatic evaluation is completely unrelated, mentioning a band's performance and the audience's reaction.

Lastly, in Figure 24, the CTN caption generated with automatic evaluation correctly links the cause of the Air Force using a surveillance system in Iraq to the effect of a woman talking about the man who created it. The caption generated without automatic evaluation is again irrelevant, discussing a group starting to dance at an event and the atmosphere becoming more lively. These examples clearly illustrate the importance of incorporating automatic evaluation in the CTN caption generation process. By ensuring that the generated captions are consistent with the video content, we can significantly improve the quality and relevance of the CTN captions, enabling the CEN model to learn more meaningful causal and temporal relationships from the videos.

### A.5 HUMAN EVALUATION STUDY DETAILS

To validate the quality of our CTN captions, we conduct a rigorous human evaluation study. This section details the methodology, results, and analysis of this evaluation.

#### A.5.1 METHODOLOGY

We employ the following methodology for our human evaluation:

- **Sample size:** 100 validation instances (50 each from MSR-VTT and MSVD datasets)
- **Evaluators:** Five independent domain experts
- **Evaluation criteria:**
    1. *Causal Accuracy:* Degree of correctly identifying and describing cause-effect relationships
    2. *Temporal Coherence:* Accuracy in representing the sequence of events
    3. *Relevance:* How well the caption reflects the overall content and context
- **Rating scale:** 5-point Likert scale (0-5, with 5 being the highest quality)

#### A.5.2 RESULTS

Table 3 presents the results of our human evaluation study.

Table 3: Human Evaluation Results for CTN Captions

| Criterion | Mean Score | Std Dev | % Perfect Scores |
|---|---|---|---|
| Causal Accuracy | 4.8 | 0.42 | 82% |
| Temporal Coherence | 4.7 | 0.46 | 78% |
| Relevance | 4.9 | 0.31 | 91% |
| Overall Quality | 4.8 | 0.40 | 84% |

Key findings from the evaluation:

- Overall mean quality score: 4.8/5.0 ($\sigma = 0.40$)

- 93% of captions received scores of 4 or higher across all dimensions

- 84% of captions achieved perfect scores

### A.5.3 INTER-RATER RELIABILITY

To ensure the consistency of ratings across evaluators, we calculate the Intraclass Correlation Coefficient (ICC) for absolute agreement among raters:

- ICC: 0.87 (95% CI: 0.83-0.91)

This high ICC value indicates strong inter-rater reliability, supporting the robustness of our evaluation process.

### A.5.4 ANALYSIS

The results of our human evaluation study strongly validate the quality and consistency of our CTN captions. With a mean quality score of 4.8/5.0 and 93% of captions receiving high scores across all dimensions, we can confidently assert that our CTN captions effectively capture causal-temporal narratives in videos. The high inter-rater reliability further strengthens the credibility of these results. These findings demonstrate that our automatic generation and evaluation process, as described in Section 3.1, produces high-quality CTN captions that accurately represent the causal and temporal dynamics in video content. This human evaluation complements our automatic evaluation metrics, providing a comprehensive validation of our CTN caption benchmark dataset.

### A.6 ENCODER COMPARISON FOR CEN

To explore the versatility of our CEN architecture, we conduct experiments using BLIP Li et al. (2022b) encoders in place of CLIP encoders. This comparison allows us to assess the adaptability of our method to different visual encoding architectures. Table 4 presents the results of this comparison on both MSVD-CTN and MSRVTT-CTN datasets.

Table 4: Comparison of CLIP and BLIP Encoders for our CEN

| Dataset | Encoder | ROUGE-L | CIDEr | SPICE |
|---------|---------|---------|-------|-------|
| MSVD-CTN | CLIP (Ours) | **31.46** | **63.51** | **19.25** |
| | BLIP | 28.79 | 57.23 | 18.06 |
| MSRVTT-CTN | CLIP (Ours) | **27.90** | **49.87** | **15.76** |
| | BLIP | 25.34 | 43.01 | 14.89 |

While the performance with CLIP encoders is higher, the results with BLIP encoders are still competitive. This demonstrates the flexibility of our CEN architecture and its potential to work effectively with various visual encoding methods.

### A.7 COMPARISON WITH NEXT-QA

To highlight the differences between our CTN benchmark dataset and the NExT-QA dataset, we provide a detailed comparison table and illustrative examples:
Table 5 presents a comprehensive comparison between NExT-QA and our NarrativeBridge approach.

Table 5: Comparison of NExT-QA and NarrativeBridge

| Aspect | NExT-QA | NarrativeBridge (Ours) |
|--------|---------|------------------------|
| Task | Question-Answering | Video Captioning |
| Output | Short answers | Coherent narratives |
| Temporal | Specific points | Multiple events |
| Causal | Implicit | Explicit |
| Narrative | Not present | Explicit |
| Focus | Info retrieval | Video description |

This comparison highlights the fundamental differences in approach and output between NExT-QA and our NarrativeBridge method.
Figure 25 provides examples from the NExT-QA dataset, illustrating the nature of its question-answer pairs. As seen in these examples, NExT-QA focuses on specific question-answer pairs that often

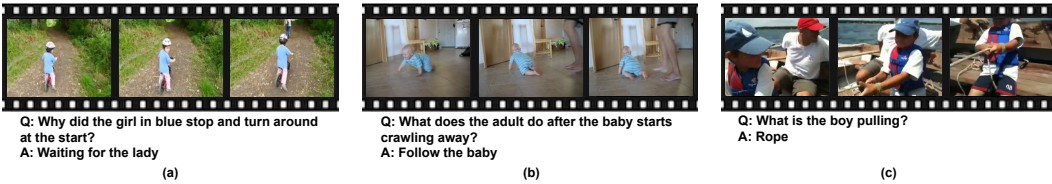

Figure 25: Examples of question-answer pairs from the NExT-QA dataset.

target single events or actions. In contrast, our CTN captions provide a more comprehensive narrative that captures the causal and temporal relationships across the entire video sequence.

## A.8 APPLICATION OF CTN CAPTION GENERATION FOR LABELING UNLABELED VIDEOS

Our CTN caption generation approach, as described in Section 3.1, can be effectively applied to the task of labeling unlabeled videos. This application leverages the power of our few-shot based prompt and the LLM's ability to generate coherent and contextually relevant captions that capture the causal-temporal narrative in video content.

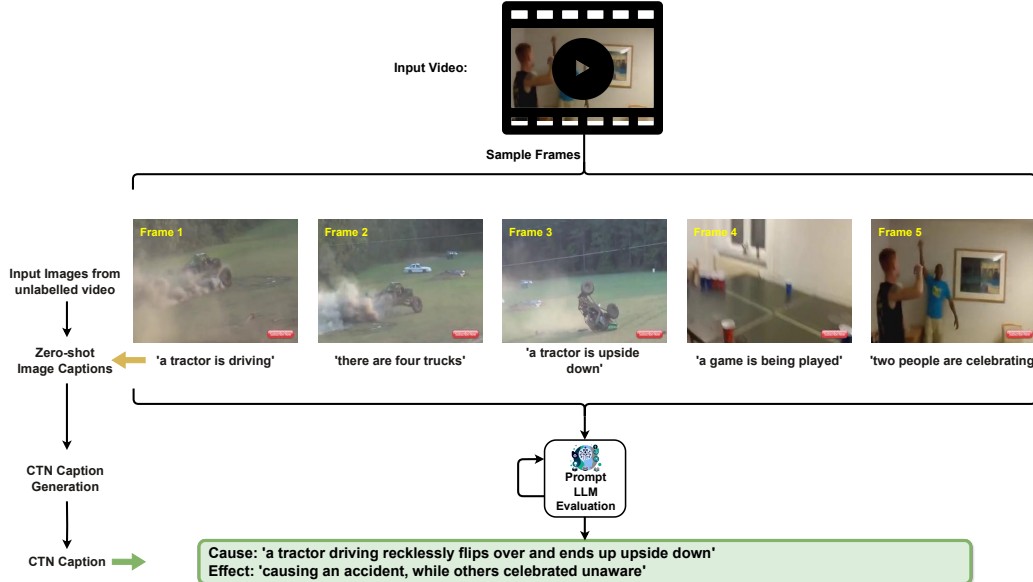

Figure 26: Application of our CTN caption generation approach for labeling unlabeled videos. Given an unlabeled video, we extract frames and generate image captions using a state-of-the-art image captioning model (GIT Wang et al. (2022)). These captions are then used as input to our LLM-based CTN caption generation pipeline, which produces a CTN caption for the entire video. The generated CTN caption captures the cause-effect relationships and temporal dynamics in the video, enabling effective labeling of unlabeled video content.

Figure 26 illustrates the pipeline for labeling an unlabeled video using our CTN caption generation approach. First, we extract frames from the unlabeled video. In this example, we extract five frames which are equally spaced in the video. Next, we generate image captions for each of the extracted frames using a state-of-the-art image captioning model. For this demonstration, we employ the GIT Wang et al. (2022), which has shown impressive performance in generating accurate captions for individual images. The GIT model generates captions such as "a tractor is driving", "there are four trucks", "a tractor is upside down", "a game is being played", and "two people are celebrating" for the five frames. These image captions serve as the input to our LLM-based CTN caption generation pipeline (see Figure 2).

We use Prompt 1 in Section 3.1, replacing the original descriptive captions with the image captions generated by the GIT model. The LLM then generates a CTN caption based on these image captions, following the specified requirements and format. In this example, the generated CTN caption is: "Cause: 'a tractor driving recklessly flips over and ends up upside down' Effect: 'causing an accident,

while others celebrated unaware'". This caption effectively captures the key events and their causal-temporal relationships in the video, providing a concise and informative summary of the video content.

To ensure the quality and relevance of the generated CTN caption, we employ the same evaluation framework described in Section 3.1. The caption is compared against the video content using the EMScore Shi et al. (2022) metric, and only captions that meet a specified threshold are retained. This application demonstrates the versatility and effectiveness of our CTN caption generation approach in labeling unlabeled videos. By leveraging the power of state-of-the-art image captioning models and our few-shot based prompt, we can generate high-quality CTN captions that accurately capture the causal-temporal narrative in video content, even in the absence of human-annotated captions. This approach has the potential to significantly streamline the process of labeling large-scale video datasets and enable more effective video understanding and retrieval tasks.

### A.9    LLM-BASED EVALUATION OF GENERATED CTN CAPTIONS

We conduct additional evaluations on MSVD-CTN and MSRVTT-CTN (combined) to evaluate further the quality of generated CTN captions, using Llama-3.2-3B-Instruct in the default configuration. Our evaluation focused on two key aspects:

**1. Temporal Order Analysis**

- Input: Ground truth and generated captions
- Task: LLM compares the temporal sequence of events
- Score: Binary (1: correct sequence, 0: incorrect)

**2. Causal Chain Analysis**

- Input: Ground truth and generated captions
- Task: LLM evaluates preservation of cause-effect relationships
- Score: Binary (1: preserved, 0: not preserved)

Table 6 shows the performance comparison between CEN and the best baseline (GIT).

Table 6: LLM-based evaluation results comparing CEN with GIT

| Model | Temporal Order (%) | Causal Chain (%) |
|---|---|---|
| CEN (Ours) | **81.2** | **84.5** |
| GIT | 52.1 | 48.3 |

We further analyze all ablations of Table 2 using the same LLM-based metrics, as shown in Table 7.

Table 7: LLM-based evaluation results for ablation study

| Method | Temporal Order (%) | Causal Chain (%) |
|---|---|---|
| CEN (Full) | **81.2** | **84.5** |
| $E_{combined}$ | 72.4 | 75.6 |
| w/o FT - single CLIP | 58.3 | 54.2 |
| w/o FT - Two CLIP | 63.7 | 61.8 |
| Only $E_{cause}$ | 73.9 | 76.5 |
| Only $E_{effect}$ | 74.8 | 77.2 |
| Zero Shot X | 55.2 | 51.7 |
| Fine-tune X | 79.4 | 82.1 |

Following are the key findings:

- CEN outperforms GIT by 29.1% in temporal and 36.2% in causal accuracy
- Dual encoder architecture provides 15% improvement over single encoder
- Fine-tuning improves temporal understanding by  20% compared to frozen models

## A.10 Additional Qualitative Results for CEN Architecture

In this section, we present additional qualitative results comparing the ground truth Causal-Temporal Narrative (CTN) captions with the captions generated by our CEN architecture. These examples further demonstrate the effectiveness of our approach in capturing the causal-temporal relationships and generating accurate and contextually relevant captions.

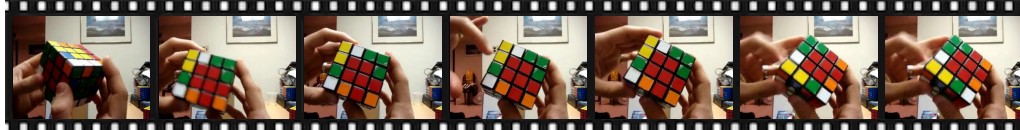

**CTN caption GT:** 'a person is working on a rubiks cube the rubiks cube gets solved piece by piece'

**CEN (OURS):** 'a person is playing with a rubiks cube and the person successfully solved the cube'

Figure 27: Comparison of CTN and CEN captions for a video of a person solving a Rubik's cube.

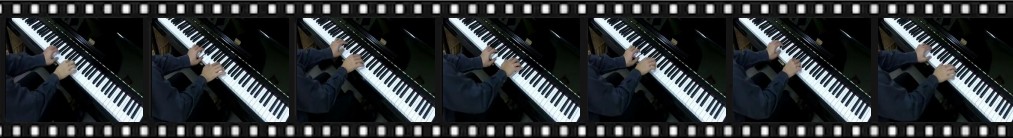

**CTN caption GT:** 'a man is playing the piano music fills the room'

**CEN (OURS):** 'the man is playing the piano and he is demonstrating how to play the piano'

Figure 28: Comparison of CTN and CEN captions for a video of a man playing the piano.

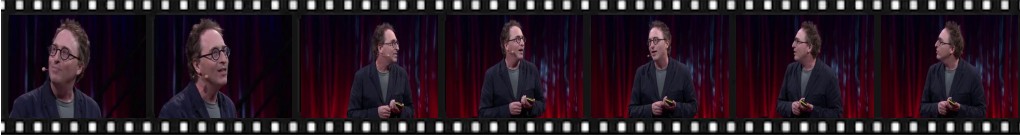

**CTN caption GT:** 'man shared a sad story on stage audience became emotionally engaged'

**CEN (OURS):** 'the man gave a lecture on stage and the audience listened intently'

Figure 29: Comparison of CTN and CEN captions for a video of a man giving a lecture on stage.

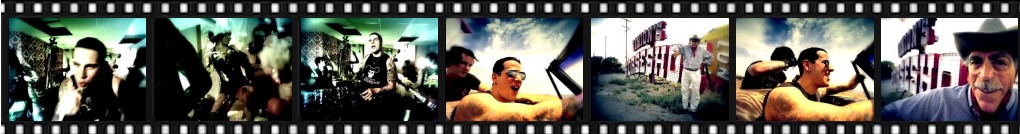

**CTN caption GT:** 'a band performs a song in a video people watch and listen to the music'

**CEN (OURS):** 'a band is performing a song and the crowd is enjoying the music'

Figure 30: Comparison of CTN and CEN captions for a video of a band performing a song.

In Figure 27, both the CTN and CEN captions accurately capture the causal relationship between the person working on the Rubik's cube and the cube being solved piece by piece. Similarly, in Figure

28, both captions correctly describe the cause-effect relationship between the man playing the piano and the music filling the room.

Figure 29 demonstrates the ability of our CEN architecture to generate captions that capture the audience's engagement in response to the man's lecture on stage. While the CTN caption specifically mentions the emotional engagement of the audience due to the sad story, the CEN caption more generally describes the audience listening intently to the lecture.

Lastly, in Figure 30, both the CTN and CEN captions accurately depict the causal relationship between the band performing a song and the crowd enjoying the music. The CEN caption, in particular, directly states the crowd's enjoyment as a result of the band's performance.

These additional qualitative examples further validate the effectiveness of our CEN architecture in understanding and articulating the causal-temporal narratives present in videos, generating captions that are coherent, accurate, and contextually relevant.

