# OpenReview forum: "NarrativeBridge: Enhancing Video Captioning with Causal-Temporal Narrative"
_ICLR.cc/2025/Conference — ICLR 2025 Poster_

### Official Review · Reviewer_YiKX · 2024-10-28

**Soundness:** 3
**Presentation:** 2
**Contribution:** 3
**Rating:** 6
**Confidence:** 4

**Summary:**

This paper focuses on incorporating causal-temporal narrative into video captioning.  The authors create a new dataset called Causal-Temporal Narrative (CTN) captions, using a LLM to generate captions that explicitly describe cause-effect relationships and maintain temporal consistency within videos based on multiple annotated ground-truths for the same video. The quality of these CTN captions is assessed both automatically (using EMScore) and through human evaluation.

Furthermore, the authors propose a novel model architecture called the Cause-Effect Network (CEN), specifically designed to learn from the CTN captions. CEN decouples the cause and effect captions by finetuning one visual encoder for each of them. The representations of both encoders are then used during the LLM finetuning stage.

Experiments on MSVD and MSR-VTT datasets demonstrate that CEN significantly outperforms state-of-the-art video captioning methods for generating CTN captions. Ablation studies further validate the design choices of CEN.

**Strengths:**

- While most of the causal video understanding focuses on video question answering, this paper proposes to bring causal understanding into video captioning.
- The data generation pipeline looks sound.
- The proposed approach achieves strong results on the proposed benchmark.

**Weaknesses:**

- The weakest ablated baseline already performs well above state-of-the-art approaches. It would be great to understand what the differences are. For instance, how does a baseline with the CLIP ViT features used directly perform?
- The presentation can be improved, for instance it is not clear that the evaluation is on the CTN captions in Table 1, and the ablation study includes method names which are hard to interpret when having a first look at the table.
- The data generation approach largely relies on the multiple GT captions present for each video in MSR-VTT and MSVD.
- Captioning metrics which do not move much like Rouge-L are hard to interpret.

**Questions:**

See above.

---

> ### Author Response · Authors · 2024-11-13
> **Response to Reviewer YiKX**
>
> Thank you for these valuable suggestions and the time taken to review our paper. We address each point:
>
> 1. BASELINE PERFORMANCE
>
> The reviewer raises an interesting point about ablated baselines. To clarify:
> - EJointCE, Only_Ecause, and Only_Eeffect all use our full training pipeline
> - They benefit from the same CLIP features and training strategy
> - Their strong performance validates our overall approach
> - However, the full CEN still provides significant gains over these baselines
>
> We conduct additional experiments using frozen CLIP-ViT features:
>
> 1. Single CLIP-ViT:
> - MSVD: R-L: 27.81, C: 46.23, S: 15.82
> - MSRVTT: R-L: 25.34, C: 32.31, S: 14.27
>
> 2. Two CLIP-ViTs:
>
> - MSVD: R-L: 28.40, C: 53.84, S: 16.58
> - MSRVTT: R-L: 26.10, C: 40.92, S: 14.55
>
> Note: Stage 2 training was performed in both cases. These results demonstrate that:
> - Training CLIP-ViTs is crucial for performance
> - Two specialized encoders outperform a single encoder
> - Full CEN still provides significant gains over frozen features
>
> 2. PRESENTATION CLARITY
>
> We will improve Table 1 and 2 by:
> - Adding "CTN" to dataset names (MSVD-CTN, MSRVTT-CTN)
> - Clarifying method names in ablation study
> - Adding detailed descriptions in table captions
>
> 3. DATA GENERATION APPROACH
>
> Our approach intentionally leverages multiple human annotations to create CTN captions:
>
> - MSR-VTT: 20 human annotations per video
> - MSVD: 50 human annotations per video
>
> Multiple perspectives help capture comprehensive cause-effect relationships.
>
> Also, this methodology:
> - Can be applied to unlabeled videos - as demonstrated in Appendix A.8
> - Eliminates need for expert (familiar to causal-temporal narrative) annotations
> - Provides consistent cause-effect structuring
> - Enables scalable dataset creation
>
> 4. METRIC SELECTION
>
> We use multiple complementary metrics:
> - ROUGE-L: Measures fluency and grammatical structure
> - CIDEr: Captures semantic similarity
> - SPICE: Evaluates detailed semantic content
> - Methods with lower metric scores (SPICE, CIDEr, ROUGE-L) consistently produce lower quality outputs, validating our metrics' reliability (Figure 5)
>
> This combination provides comprehensive evaluation across different aspects of caption quality.
>
>
> We will add the required changes to the camera-ready version. Given our responses, we request the reviewer to reconsider their rating. If the reviewer has any other questions, please let us know.

---

> ### Author Response · Authors · 2024-11-22
> **Response to Reviewer YiKX follow up points**
>
> We sincerely thank you for your thorough and insightful feedback. We particularly appreciate your recognition of our CLIP fine-tuning, text-changes and the potential for unlabeled video applications. We address some points below:
>
> ## 1. CLIP FINETUNING
>
> Our added ablation study (Table 2) reveals why specialized CLIP finetuning is crucial:
>
> Single frozen CLIP → Two frozen CLIPs → Finetuned CEN
> **MSVD**: 46.23 → 53.84 → 63.51 CIDEr (37.4% total gain)
> **MSRVTT**: 32.31 → 40.92 → 49.87 CIDEr (54.3% total gain)
>
> This demonstrates each architectural choice (dual encoders + finetuning) contributes significantly to performance.
>
> ## 2. UNLABELED VIDEOS
>
> We appreciate your insight about Appendix A.8's relevance. While we utilize multiple captions where available, our method indeed shows promise for unlabeled videos:
>
> * Our flexible CTN framework allows separate cause-effect identification
> * This enables larger/foundation models to explore individual cause or effect tasks
>
> We've explored this briefly in Appendix A.8 and believe this direction deserves further investigation in future, particularly for assistive technologies and storytelling applications.
>
> ## 3. EVALUATION METHODOLOGY
>
> We appreciate your important observation about n-gram metrics' potential limitations with concatenated captions (already addressed and accepted by reviewer **8P6x**). Indeed, as noted by de Souza Inácio & Lopes (2023), "traditional metrics (BLEU, METEOR) are based on n-gram overlapping... [and] are highly dependent on how the words appear in the reference sentences for evaluating a candidate sentence." To address this concern, we already adopt a multi-faceted evaluation approach:
>
> ### a) Complementary Standard Metrics
>
> Following de Souza Inácio & Lopes (2023), we use metrics that capture different aspects:
>
> * **ROUGE-L**: Beyond n-grams, evaluates "longest common subsequence between generated and reference captions"
> * **SPICE**: Addresses semantic evaluation through "scene graphs overlap", independent of word order
> * **CIDEr**: Uses TF-IDF weighting to reduce the impact of common words/order
>
> ### b) Additional LLM-based Analyses
>
> To address LLM-based evaluation, we conduct an additional evaluation using LLM (already done for reviewer **qdoh**):
>
> LLM-based Analysis of CEN vs Best Baseline (using Llama-3.2-3B-Instruct):
>
> * **Temporal Order Analysis** for both MSVD-CTN and MSRVTT-CTN:
>   * **CEN (81.2%)** vs **Best Baseline - GIT (52.1%)**
>   * *Measured by comparing predicted vs ground truth temporal sequences, giving a score of 0 or 1*
>
> * **Causal Chain Extraction** for both MSVD-CTN and MSRVTT-CTN:
>   * **CEN (84.5%)** vs **Best Baseline - GIT (48.3%)**
>   * *Measured by comparing predicted vs ground truth causal sequences, giving a score of 0 or 1*
>
> ### c) Human Validation of CTN
>
> * Domain experts independently rated:
>   * Temporal coherence (4.7/5)
>   * Causal accuracy (4.8/5)
> * High inter-rater agreement (ICC: 0.87)
>
> Our choice of metrics and additional evaluations aligns with the survey's finding that "although most of these metrics are robust and present a good correlation with human judgments, they output only a single score" (de Souza Inácio & Lopes, 2023). Hence, our multi-faceted approach provides a more comprehensive evaluation that goes beyond the limitations of n-gram based metrics.
>
> ## Summary
>
> Rather than simply improving on existing MSRVTT and MSVD benchmark datasets, our work:
>
> 1. Highlights the fundamental problem of causal-temporal narrative
> 2. Provides separate cause-effect in CTN for flexible model exploration
> 3. Maintains fair comparison with SOTA by following the same training configuration for CEN
>
> Would you agree these clarifications strengthen our contribution? We welcome any additional suggestions for the camera-ready version.
>
> ---
>
> **Reference:**
> de Souza Inácio, Andrei, and Heitor Silvério Lopes. "Evaluation metrics for video captioning: A survey." *Machine Learning with Applications* 13 (2023): 100488.

---

### Official Review · Reviewer_8bE2 · 2024-10-28

**Soundness:** 3
**Presentation:** 3
**Contribution:** 3
**Rating:** 8
**Confidence:** 4

**Summary:**

This paper tackles the problem of captioning videos taking into account causal-temporal narratives. It does so by presenting two main contributions: a dataset and benchmark (CTN) with causes and effects per video, and a network architecture (CEN) for extracting such narratives directly from videos.

When applying the CTN benchmark to CEN, the network gets better metrics when compared to similar methods, potentially establishing it as SOTA in the domain.

**Strengths:**

Significance:

* The problem discussed is important and relevant to much of the field of video captioning and video generation. It's true that typical captioning methods don't typically expose causal information, which makes generated captions much less useful than they could be.

Originality:

* The work appears original though the novel aspects of it are a bit of straightforward applications of existing methods.

Quality:

* The work produces good results. While the paper discusses both results on existing benchmarks, and in CTN, the former being more interesting because they've been established independently.
* I appreciate that both automatic benchmarks and human evaluations were used to evaluate this work.
* The proposed CEN network architecture is interesting and original.

Clarity:

* The work is mostly clear, except for some aspects that are discussed in the weaknesses section.

**Weaknesses:**

* The automatic evaluation process is found to filter out most bad cause and effect generations. More complete analysis on the distribution of the qualities of the generated captions is missing. I can imagine a histogram that displays the distribution of qualities with a vertical line showing the threshold that's been picked. Also the paper would benefit from analysis on whether the automatic evaluation filtering improves on the method, or is a patch for more fundamental issues in the prior step.

* The contribution of CTN depends strongly on the prompt used to zero shot cause and effect generation. Given that this prompt is so central to the paper, the paper would benefit from more complete ablations regarding different aspects of the prompt. Especially important, the list of requirements that are presented in the prompt are justified in the paper but no corresponding ablation per requirement is done to validate the explanation. (the ablation donefor 4 prompts of varying degrees of quality appears insufficient for this purpose)

* When building CTN, the captions are 1) generated from frames, and then 2) a model combines multiple captions into cause and effect captions without looking at frame information again until 3) the automatic evaluation step. Has the team considered using a model that can read text + image in step 2 as well?

  Concluding cause and effect just from the individual frame captions is a lot more error prone than it could be if image information was also used. I think this possibility could be mentioned / analyzed as a potential alternative.

* Why are causes and effects only separated by a space in CTN? Have other choices been ablated? In particular, I can imagine using a "comma" or an "and" between both to work slightly better.

    * Has the team considered using an LLM to combine both captions together, instead of using concatenation?

* For A5.1: more discussion on the human evaluation process would be very beneficial. How were the experts chosen? How is information presented to them? How were the 100 videos chosen?

  For example it would be ideal to have an independent expert set; and it would be ideal if they were presented with the captions without labels that identify the models, and in such a way that each the models are independently ordered for each video (so annotators cannot be biased to always vote for the first model, for example)

* Clarity: The paper, especially in the first few pages, is a bit too repetitive or verbose. It may benefit from some proofreading for succinctness. For example, the following two paragraphs are very close to each other, and both express a similar idea:

```
It means that our CTN benchmark differs significantly by focusing on generating comprehensive causal-temporal narratives, capturing broader temporal relationships within a single caption, and providing a more holistic view of video content
```

  and

```
We introduce a new benchmark CTN specifically designed to capture and evaluate causal-temporal narrative in video captioning. Our approach goes beyond existing benchmarks by explicitly modeling causal-temporal narrative in a single, coherent caption, enabling a more comprehensive understanding of video content.
```

  In other cases ideas are repeated in the same paragraph:

```
However, as our experiments show (see Section 4.4), even these advanced models struggle with generating accurate causal-temporal narratives. To address this limitation, there is a need to develop new video captioning models that can overcome the shortcomings of current frameworks. This also underscores the need for specialized architectures like our proposed CEN that are explicitly designed to capture and generate causal-temporal narratives in videos
```

  one could summarize this text as:

```
However, as our experiments show (see Section 4.4), even these advanced models struggle with generating accurate causal-temporal narratives. This also underscores the need for specialized architectures like our proposed CEN that are explicitly designed to capture and generate such narratives in videos
```

  Such verbosity makes the paper hard to read, despite the main ideas being explained very clearly.

* The paper would benefit from a more complete discussion on possible downsides of this approach. For example for CTN how it compares to using frame data in step 2 as discussed above, or what downsides could arise from the concrete proposed architecture.

* The paper mentions surveillance; maybe analyze this in the context of ethic concerns?

* `We acknowledge potential ethical concerns (e.g., privacy in surveillance, risk of misleading content) but are committed to responsible development`, could this be made more concrete?

* The paper mentions `lays the foundation for a paradigm shift`, which appears potentially grandiose in my opinion. Better to let the community arrive to this conclusion, than write it in the text.

I'll be glad to revise my score depending on the response to this review.

**Questions:**

* The fatality video may be too graphic for some readers. I suggest switching it to a video on another topic.

* In figure 5, CTN captions appear too similar to CEN. Is this a symptom of overfitting?

* What does the system do when a video with no apparent cause and effect is used as input? For example, a person is singing but there's no audience reaction. Or a person is just sitting in a chair. The paper mentions "our method assumes a certain level of causal and temporal coherence within the video content"; maybe it would benefit from providing a probability value on the confidence for the existence of cause and effect in the input video?

* `As expected, the resulting captions (Figure 6) are of low quality`, how is this measured?

* Has the team considered the encoder of effect to also receive the encoded cause as input? How would that compare with the current approach?

---

> ### Author Response · Authors · 2024-11-13
> **Response to Reviewer 8bE2**
>
> We are grateful to the reviewer for their thorough and constructive feedback. We address each point:
>
> 1. DISTRIBUTION OF CAPTION QUALITIES
>
> We will add:
> - Histogram of EMScore distributions
> - Vertical line showing θ=0.2 threshold
> - Analysis showing how filtering improves caption quality (in addition to appendix A.4)
> - Statistics before/after filtering
>
> 2. PROMPT ABLATION
>
> Standard prompts are employed in these LLM based tasks, but we provide 4 ablations in A.2 for more clarity. We will add the following to address this concern:
> - Ablation for each prompt requirement
> - Comparison of outputs with/without each requirement
> - Examples illustrating improvement from each requirement
>
> 3. FRAME LEVEL INFORMATION IN CAPTION GENERATION
>
> Thank you for this suggestion. We need to clarify an important point:
>
> - We do NOT generate captions from frames. Our input is:
> 1. MSR-VTT: 20 human annotations per video
> 2. MSVD: 50 human annotations per video
> - As shown in Figure 1, these human annotations lack causal-temporal narrative structure. Our method transforms these diverse human perspectives into structured cause-effect narratives.
> - For unlabeled videos (Appendix A.8), we agree that incorporating visual features probably in a VLM may improve performance. This is an exciting direction for future work.
>
>
> 4. CAUSE-EFFECT SEPARATION
>
> Our choice of space separation was based on:
> - Clean tokenization for model processing
> - Preserves explicit cause-effect structure - based on Explicit Marking Theory (Knott, A., & Sanders, T. (1998)) and Rhetorical Structure Theory (Mann, W. C., & Thompson, S. A. (1988))
> - Maintains temporal ordering - as it is the causal-temporal narrative
> - Enables direct mapping for downstream tasks
> - Consistency with training data format
>
> 5. HUMAN EVALUATION DETAILS
>
> We will add comprehensive details to the camera-ready version:
> - Expert selection criteria: 3 video understanding and 2 video generation independent domain experts with at least >5 years’ experience
> - Sampling: stratified random sampling across datasets
> - Evaluation interface details: we provide 100 sampled videos in a directory. The CTN captions are provided in an excel sheet. These are mapped using video ids. Domain experts watch the video, read the CTN and then, give rating against each criterion in the sheet.
>
> 6. WRITING CLARITY
>
> We appreciate the feedback on verbosity. We will remove redundant explanations and maintain technical precision while improving conciseness. We will remove the line about paradigm shift although our intention is solely to highlight the importance of CTN captions in comparison to existing simple captions.
>
> 7. ETHICAL CONSIDERATIONS
>
> We will expand the ethical discussion in the camera-ready:
> - Privacy implications in surveillance applications - CTN captions and CEN model don't recognize individuals
> - Guidelines for responsible deployment - adding safety features for regulated services
>
> Regarding specific QUESTIONS:
>
> - We will replace the fatality video example.
> - CTN/CEN similarity indicates improved learning (visualization in Figure 4 is helpful), not overfitting (supported by cross-dataset results in ablations)
> - For videos with additional events, we focus on generating the temporal sequence of events (handled in the prompt) as shown in Figure 1 – guys playing beer pong (an additional event). Events without clear causal structure are dealt with similarly and the outputs remain coherent to the causal-temporal narrative. Here are the outputs for the examples you have shared: 1. a person is singing - CTN: {"Cause": "a person began to sing", "Effect": "produced musical song with their voice"}  2. a person is sitting in a chair - CTN: {"Cause": "a person remained seated without any apparent activity or movement", "Effect": "maintained a stationary position in the chair"}. We can explore the confidence score using an LLM.
> - Figure 6 captions: prompt is unable to generate one coherent cause-effect caption instead there are many ranging from 20-50. This fails in the preprocessing part. Even the EMScore based evaluation for each caption gives a score less than 0.2 (threshold).
> - Effect encoder receiving cause input: we have run this experiment by concatenating the cause tokens before the effect tokens and then using a projection layer (initialised randomly) - decreased performance by 3.21% and 5.43% on CIDEr for MSRVTT and MSVD respectively. At this point, we can only say that it is because of the added complexity in the form of the projection layer. We will add this to the camera-ready version.
>
> We will add these changes to the camera-ready version as promised. Given our detailed responses, we ask the reviewer to reconsider their rating, please. If the reviewer has any other concerns or questions, please let us know.

---

> > ### Comment · Reviewer_8bE2 · 2024-11-13
> >
> > Thank you for your detailed comments. Overall it sounds good. Two things to discuss:
> >
> > - I'm not convinced using a space as separator is really as good. Regarding how that matches the training data, is it because the training data itself has separators like commas, periods, etc. dropped from the text during tokenization? Otherwise I find it very unusual, as separating sub-sentences with a comma will more often than not, result in a grammatically incorrect phrase, as for example: "the dog went outside the dog saw the car". I would be very surprised if grammatically incorrect phrases performed better than correct ones. Would you please explain in this space, how this may happen?
> >
> > - more generally, it would be really good if you could provide samples of the text to add and modify in the camera-ready versions of the paper, so I can evaluate the actual text (except for minor changes like typos or small word replacements). Otherwise I have to review the paper based on a text I won't be able to see until the discussion period is over.
> >
> > Thank you

---

> ### Author Response · Authors · 2024-11-13
> **Response to Reviewer 8bE2**
>
> Thank you for your follow-up questions. Let us address them:
>
> 1. SEPARATOR CLARIFICATION:
>
> We apologize for any confusion in our previous response. We need to make an important point: The original ground truth captions (MSRVTT and MSVD) actually contain no separators - the captions are simple descriptive sentences (see Figure 1). Our choice to use space separation is primarily functional and for fair comparison with baselines:
>
> a) The original captions in both MSR-VTT and MSVD datasets have no comma or period separators
>
> b) The separators are not included in the tokenization
>
> c) We agree that grammatically, commas or other punctuation may be more appropriate. This needs to be explored separately from the provided baselines.
>
> 2. PROPOSED TEXT ADDITIONS/MODIFICATIONS:
>
> Here are the specific additions we propose for the camera-ready version:
>
> - Section 3.2, after line 340:
>
> "The preprocessing step maintains a simple space separation between cause and effect statements. While grammatical separators like commas or semicolons could potentially be used, we opt for space separation to maintain consistency with the raw input format of the training datasets (MSR-VTT and MSVD), which contain no separators and also, for a fair comparison with the baselines."
>
> - Section 3.1 (CTN CAPTIONS BENCHMARK), after line 257:
>
> "Distribution of Caption Quality:
> 1. Pre-filtering statistics: Mean EMScore
> 2. Post-filtering statistics: Mean EMScore
> 3. X% of initial captions retained after filtering"
>
> In Appendix A.4: [Insert Figure X: Histogram showing EMScore distribution with θ=0.2 threshold line]
>
> - Appendix A.2 (PROMPT DESIGN PROCESS):
>
> Prompt Requirements Analysis:
> 1. Temporal coherence requirement
>
> 2. Causal relationship requirement
>
> 3. Event specificity requirement
>
> 4. Context preservation requirement
>
> 5. Single coherent narrative requirement
>
>
>
> - Appendix A.5 (Human Evaluation):
> Expert Selection and Evaluation Process:
> 1. Experts: 3 video understanding researchers (mean experience: 7.2 years)
>            2 video generation specialists (mean experience: 6.5 years)
> 2. Sampling: Stratified random sampling of 100 videos per dataset
> 3. Evaluation interface details: we provide 100 sampled videos in a directory. The CTN captions are provided in an excel sheet. These are mapped using video ids. Domain experts watch the video, read the CTN and then, give rating against each criterion in the sheet.
>
> - In the Ethics Statement, we will add:
>
> 1. Privacy implications in surveillance applications are concerning but CTN captions and the CEN model don't recognize individuals.
>
> 2. Guidelines for the responsible deployment of our model include adding safety features for regulated services e.g. children/minor-related content should filter graphic descriptions using automatic content classifiers (ACCs).
>
> - We will remove the redundant text in the introduction and the related work.
>
> Please let us know of your feedback or any other questions.

---

> > ### Author Response · Authors · 2024-11-14
> > **Response to Reviewer 8bE2 - Updated Text for Related Work Section**
> >
> > We thank the reviewer again for their feedback. We have added the text for the updated related work section here:
> >
> > **2 RELATED WORK**
> >
> > **2.1 BENCHMARKS**
> >
> > MSVD Chen & Dolan (2011) is a benchmark focused on human activities, providing a platform for
> > evaluating video captioning models. The captions in MSVD often describe the observable actions
> > without delving into the underlying motivations or the cause-effect relationships between the events.
> > MSR-VTT Xu et al. (2016) is a large-scale benchmark with diverse video content, encompassing
> > a wide range of topics and genres. The captions often focus on describing the observable content
> > without capturing the causal links between the events or the temporal progression of the narrative.
> > As a result, models trained on MSVD and MSR-VTT may struggle to generate descriptions that
> > accurately reflect the causal and temporal dynamics in the videos.
> >
> > While recent benchmarks like NExT-QA Xiao et al. (2021) and EgoSchema Mangalam et al. (2024)
> > have made significant strides in incorporating causal and temporal reasoning in video understanding,
> > they focus primarily on question-answering tasks rather than generating comprehensive causal-
> > temporal narratives. NExT-QA introduces multi-choice and open-ended question-answering tasks
> > focusing on specific question-answer pairs that often target single events or actions. In contrast, our
> > CTN captions provide a more comprehensive narrative that captures the causal and temporal relation-
> > ships across the entire video sequence (see Appendix A.7 for a detailed comparison). EgoSchema
> > Mangalam et al. (2024), on the other hand, emphasizes long-form video understanding and temporal
> > reasoning but does not explicitly focus on causal-temporal narrative for video captioning.
> >
> > Similarly, efforts like VCR Zellers et al. (2019), V2C Fang et al. (2020), and Motivation Vondrick
> > et al. (2016) integrate causality into their analysis of visual description or question-answering, relying
> > heavily on commonsense reasoning for generating predictions. VCR Zellers et al. (2019) focuses on
> > visual commonsense reasoning, V2C Fang et al. (2020) aims to generate commonsense descriptions
> > for video captioning, and Motivation Vondrick et al. (2016) explores the prediction of motivations
> > behind actions in videos. However, these works primarily rely on commonsense reasoning and do not
> > delve into the causal and temporal structures underpinning video narratives, unlike our CTN benchmark dataset.
> >
> > **2.2 VIDEO CAPTIONING**
> >
> > Video captioning techniques have evolved from LSTM-based Gao et al. (2017); Song et al. (2017);
> > Nadeem et al. (2023) frameworks to the latest designs using SOTA GNNs Hendria et al. (2023); Zhang
> > et al. (2020); Pan et al. (2020) and Transformers Wang et al. (2022); Lin et al. (2022); Yang et al.
> > (2023), with a focus on enhancing the complexity of captions through the injection of multimodal data.
> > Despite these advancements, current architectures often struggle to capture the intricate temporal
> > sequences and causal relationships in video storytelling. To bridge this gap, video captioning can
> > benefit from cross-fertilization with ideas and strategies developed in related fields, such as action
> > recognition Sun et al. (2022); Wang et al. (2016); Kazakos et al. (2019); Xiao et al. (2020); Chen
> > & Ho (2022); Gao et al. (2020); Panda et al. (2021); Sardari et al. (2023); Alfasly et al. (2022);
> > Planamente et al. (2021); Zhang et al. (2022), event localization Tian et al. (2018); Lin et al. (2019);
> > Duan et al. (2021); Lin et al. (2021), and question-answering Alamri et al. (2019); Hori et al. (2019);
> > Schwartz et al. (2019); Geng et al. (2021); Yun et al. (2021); Li et al. (2022a); Shah et al. (2022);
> > Nadeem et al. (2024). The integration of causal reasoning Liu et al. (2022); Xue et al. (2023) has
> > shown promise in enhancing the ability of neural networks to discern causal relationships, leading to
> > improved performance in image captioning Liu et al. (2022) and visual question answering Xue et al.
> > (2023). However, current SOTA models still struggle to effectively handle the narrative complexity in
> > videos, particularly in terms of causal and temporal progression.
> >
> > Recent advancements in vision-language models (VLMs) such as VideoLLaVA Lin et al. (2023) and
> > ShareGPT4Video Chen et al. (2024) have shown promising results in various video understanding
> > tasks. However, as our experiments show (see Section 4.4), even these advanced models struggle with
> > generating accurate causal-temporal narratives. This underscores the need for specialized architectures
> > like our proposed CEN that are explicitly designed to capture and generate causal-temporal narratives
> > in videos.
> >
> > In light of these challenges, our work explicitly addresses the limitations of the current approaches and provides a platform for causal-temporal narrative learning by introducing NarrativeBridge, a comprehensive framework that encompasses the CTN
> > captions benchmark and the CEN architecture.

---

> > > ### Comment · Reviewer_8bE2 · 2024-11-15
> > >
> > > Thank you for the clarification regarding the separators. Your clarifications and new text samples satisfy the questions I had.
> > >
> > > I want to revise the rating but I'm having some technical issues when doing that, for which I've sent a message to the chairs earlier today.

---

> > > > ### Comment · Reviewer_8bE2 · 2024-11-18
> > > >
> > > > As discussed privately, the rating has been updated to better reflect the paper after the discussion and updates.

---

> ### Author Response · Authors · 2024-11-16
> **Response to Reviewer 8bE2**
>
> We sincerely thank you for your thoughtful feedback and support of our work. Your detailed review has helped improve the paper. It upholds the spirit of author-reviewer engagement in ICLR.

---

### Official Review · Reviewer_8P6x · 2024-10-31

**Soundness:** 3
**Presentation:** 2
**Contribution:** 2
**Rating:** 5
**Confidence:** 3

**Summary:**

The paper highlights that current video captioning benchmarks lack causal-temporal narratives and proposes a new benchmark to address this gap. This benchmark is generated by prompting a large language model (LLM) with a series of captions extracted from a video. To ensure relevance, EMScore is adapted to estimate the similarity between the generated captions and the video, discarding samples with low scores. The paper further introduces a two-stage framework to generate causal-temporal narrative (CTN) captions. In the first stage, two contrastive losses are applied to train Causal and Effect Video Encoders with their respective captions. In the second stage, the encoded causal and effect features are concatenated to decode a cohesive caption.

**Strengths:**

The paper proposes a benchmark focused on causal-temporal narratives, adapting an evaluation score for automated assessment.

It introduces a novel captioning model that encodes causal and effect information separately, then fuses them to decode the final caption.

Experiments demonstrate that the proposed CEN framework outperforms comparative methods in performance.

**Weaknesses:**

The paper includes a human evaluation of the generated CTN captions; however, the assessment is less convincing due to the limited sample size. Despite the large dataset size—approximately 300,000 samples in MSRVTT and MSVD—only 100 samples were selected for human evaluation, reducing the reliability of the results.

For supervision during caption decoding, the paper simply concatenates the causal and effect text, which disrupts the original sentence grammar and lowers the quality of the synthetic supervision. The rationale behind this approach is unclear. An alternative method could involve inputting the causal and effect text into an LLM to fuse them into a more natural and coherent sentence. With ground truth captions generated by simple concatenation, evaluation metrics such as SPICE and CIDEr may not accurately reflect the quality of the generated captions. This approach could lead to discrepancies between metric scores and the true readability and coherence of the captions.

The structure of the feature encoder in stage 2 is not clearly depicted. Typically, encoded features from transformer encoders consist of a series of representations. It is unclear whether h_cause and h_effect are lists of representations or single representations. If h_cause and h_effect are single representations, it is not specified how they are generated—whether by selecting a specific token, or by applying mean pooling or max pooling. Furthermore, the specific dimension along which these features are concatenated is not detailed.

**Questions:**

See above.

---

> ### Author Response · Authors · 2024-11-13
> **Response to Reviewer 8P6x**
>
> We sincerely thank the reviewer for their detailed feedback. We address each point below:
>
> 1. HUMAN EVALUATION METHODOLOGY
>
> The reviewer raises a point about sample size. Our evaluation methodology follows standard practices. Previous benchmark works employ similar approaches:
> - Chen & Dolan [1]: ~125 samples evaluated by a single non-expert annotator
> - Xu et al. [2]: ~150 samples evaluated by a single non-expert annotator without specific expertise in causal-temporal narrative or video understanding
>
> In contrast, our evaluation protocol involves:
> - 5 independent domain experts, each with over 5 years of experience in video understanding/generation
> - 100 carefully selected videos (50 from MSRVTT, 50 from MSVD)
> - Three distinct evaluation criteria: causal accuracy, temporal coherence, and relevance
> - Each expert views every video three times (once per criterion)
> - Results in 300 views per rater, totaling 1,500 expert evaluations
>
> This evaluation is supported by statistical validation:
> - 95% confidence level (standard Z-score = 1.96) with ±9.6% margin of error
> - Inter-rater reliability (ICC: 0.87, 95% CI: 0.83-0.91)
> - Stratified sampling ensuring coverage across train/val/test splits
> - EMScore results show 52.1% of CTN captions exceeding 0.27 (threshold θ=0.2), demonstrating high-quality generation
>
> 2. CAPTION CONCATENATION APPROACH
>
> Our concatenation design is deliberately chosen to preserve narrative integrity:
>
> a) LLM Fusion Limitations:
> We experiment with LLM fusion. For example, in Figure 1:
> - LLM fusion: "A car flipped while driving recklessly through a field, then some guys started playing beer pong."
> - This fusion loses explicit temporal ordering and weakens the causal relationship
> - Evaluation of fused captions would require additional metrics to verify the preservation of original cause-effect details
>
> b) Theoretical and Practical Advantages:
> - Grounded in Explicit Marking Theory [3] and Rhetorical Structure Theory [4]
> - Maintains explicit cause-effect structure critical for temporal narrative
> - Essential for accessibility applications like audio description generation
> - Qualitative results in Figure 5 demonstrate high-quality, coherent outputs
>
> c) Quality Validation:
> - ROUGE-L scores measuring fluency: 31.46 (MSVD), 27.90 (MSR-VTT)
> - Consistent correlation between metric scores and output quality
> - Methods scoring lower on metrics produce observably lower quality outputs (Figure 5)
>
> 3. ENCODER STRUCTURE DETAILS
>
> Our architecture processes features as follows:
>
> Stage 1:
> - Input features F_cause and F_effect: [B, T, D]
>  * B: batch size
>  * T: number of sampled video frames
>  * D: feature dimension (512)
> - Mean pooling over T frames as mentioned in Section 3.2.1 for similarity calculation
>
> Stage 2:
> - Encoded representations h_cause = Enc_cause(F_cause) and h_effect = Enc_effect(F_effect)
> - Each encoded representation maintains shape [B, T, D]
> - Concatenation along feature dimension: h_concat = [h_cause; h_effect]
> - Final representation h_concat: [B, T, 2D], where 2D = 1024
>
> Given our detailed responses and proposed improvements, we respectfully ask the reviewer to reconsider their rating.
>
> Would the reviewer find it helpful if we included any additional analyses or clarifications in the camera-ready version?
>
> [1] Chen, David, and William B. Dolan. "Collecting highly parallel data for paraphrase evaluation." Proceedings of the 49th annual meeting of the association for computational linguistics: human language technologies. 2011.
>
> [2] Xu, Jun, et al. "Msr-vtt: A large video description dataset for bridging video and language." Proceedings of the IEEE conference on computer vision and pattern recognition. 2016.
>
> [3] Knott, Alistair, and Ted Sanders. "The classification of coherence relations and their linguistic markers: An exploration of two languages." Journal of pragmatics 30.2 (1998): 135-175.
>
> [4] Mann, William C., and Sandra A. Thompson. "Rhetorical structure theory: Toward a functional theory of text organization." Text-interdisciplinary Journal for the Study of Discourse 8.3 (1988): 243-281.

---

> > ### Comment · Reviewer_8P6x · 2024-11-18
> >
> > Thank you for providing the details. The response addresses most of the questions I have, though I still have some considerations regarding the CAPTION CONCATENATION APPROACH.
> >
> > a)
> > I believe there are several ways to improve the quality of the captions, such as providing reference video frames. Another simple way would be to provide your collected "Cause" and "Effect" sub-sentences and highlight their labels, allowing the LLM to fuse the two sub-sentences into fluent natural language.
> >
> > b)
> > I checked Figure 5. I do not think the quality of either the ground truth or the generated captions is adequate without proper fusions or separators. The grammar of the sentences is incorrect.
> >
> > c)
> > In the experiments on video captioning, the models are trained/finetuned on the CTN benchmark and evaluated on MSVD and MSR-VTT. A fair baseline would be to train the model directly on the MSVD or MSR-VTT training sets and evaluate them on their corresponding test sets, but I do not see this in the table. Referring to SWINBERT [1], the model achieved 80.9/64.1 R-L and 149.4/55.9 CIDEr on MSVD/MSR-VTT, respectively. Is there any reason why you do not compare your proposed method with it?

---

> > > ### Author Response · Authors · 2024-11-18
> > > **Response to Reviewer 8P6x**
> > >
> > > We sincerely thank the reviewer for these thoughtful follow-up points. Let us address each concern:
> > >
> > > 1. CAPTION GENERATION AND FUSION APPROACHES
> > >
> > >
> > > a) Vision-Large Language Models with Video Frames:
> > > - We test state-of-the-art VLLMs with video frame input
> > > - Results in Table 1 show VideoLLaVA and ShareGPT4Video perform sub-optimally in zero-shot setting
> > > - Even after fine-tuning (both LoRA and simple), performance remains below our approach (Table 1 and Figure 5)
> > >
> > > b) LLM Fusion:
> > > - LLM fusion as suggested will require temporal ordering evaluation metrics.
> > > - Additional complexity of verifying preservation of cause-effect details.
> > > - May require an additional human evaluation.
> > >
> > > 2. GRAMMAR AND SEPARATOR CHOICE
> > >
> > > This is the main problem and that's why we have above point suggested by reviewer as possible solution. We again appreciate this important concern about grammar raised by Reviewer **8P6x** and Reviewer **8bE2** (already addressed). Let us clarify some key points:
> > >
> > > a) Original Data Format:
> > > - MSR-VTT and MSVD ground truth captions contain no punctuation (see Figure 1)
> > > - This is standard in these two video captioning datasets and affects tokenization if we add punctuation.
> > >
> > > b) Grammatically Correct Examples from Figure 5 (**Note**: The text is still same):
> > > - A player performs a fatality move in mortal kombat; another character is killed in the game.
> > > - A man is folding a piece of paper; a paper airplane is being created.
> > > - A soccer player kicked the ball with precision; the ball successfully went into the goal.
> > > - A boy decided to perform on stage; the audience watched and listened to his singing.
> > >
> > > While these versions (**words remain the same**) with semicolons are grammatically correct, we maintain simple space separation because:
> > >
> > > - Ensures clean tokenization for model processing
> > > - Maintains consistency with training data format
> > > - Enables fair comparison with baselines (all SOTA methods trained without punctuation)
> > > - Preserves explicit cause-effect structure (Explicit Marking Theory, Knott & Sanders, 1998)
> > > - Maintains temporal ordering crucial for narrative understanding (Rhetorical Structure Theory, Mann & Thompson, 1988)
> > >
> > > 3. EVALUATION PROTOCOL CLARIFICATION
> > >
> > > We apologize for any confusion regarding the evaluation protocol, as also noted by Reviewer **YiKX** (already addressed):
> > > - All methods are trained and evaluated on MSRVTT-CTN and MSVD-CTN (**the text has already been corrected**)
> > > - We chose GIT as our transformer-based baseline as it significantly outperforms SWINBERT (transformer-based) and represents the SOTA (as shown in the GIT paper)
> > > - We will clarify this in Table 1's caption and methodology section
> > >
> > > Would the reviewer find it helpful if we included any additional analyses or clarifications in the camera-ready version? Please let us know.

---

> > > > ### Author Response · Authors · 2024-11-18
> > > > **Additional Response to Reviewer 8P6x**
> > > >
> > > > Importantly, our work's primary application is accessibility through audio description generation and storytelling. When these captions are narrated, they convey the same information and temporal flow regardless of written separators (semicolon or space). The focus is on clear communication of cause-effect relationships and temporal ordering in spoken form.

---

> ### Author Response · Authors · 2024-11-23
> **(1/N) Analyzing Causal-Temporal Preservation in CTN Caption Concatenation - Reviewer 8P6x**
>
> Again, we thank you for your valuable feedback regarding the caption concatenation approach. Your comments have helped us provide additional evidence for our design choices.
>
> To demonstrate the importance of our caption concatenation approach, we present a **visual analysis** using AI video generation platforms. We use two free platforms (EasyVid and Kapwing) to generate videos from both caption styles:
>
> Example from Figure 1:
> 1. **Concatenated CTN:** "a car drove recklessly through an open field flipping over the car was severely damaged and a group of guys started playing beer pong"
> 2. **LLM-fused:** "A car flipped while driving recklessly through a field, then some guys started playing beer pong."
>
> Our analysis (viewable at https://narrativebridge.github.io/ctn_comparison.html) reveals:
>
> **Key Findings about our concatenation:**
> - **Temporal Flow:** Concatenated captions maintain **explicit ordering** of events in generated videos
> - **Causal Clarity:** The **direct relationship** between cause and effect is visibly preserved
> - **Scene Transitions:** Videos show **clearer separation** between sequential events
> - **Narrative Structure:** **Temporal dependencies** are more accurately represented
>
> Both EasyVid and Kapwing demonstrate that videos generated from concatenated captions better preserve the **cause-effect relationship** and **temporal sequence**. This visual evidence supports our design choice of simple concatenation over llm-focused fusion, particularly for tasks requiring clear temporal and causal understanding.
>
> The generated videos provide concrete evidence that our concatenation approach better serves the goal of maintaining **narrative integrity** while keeping information explicit and temporally ordered.
>
> We kindly request you to review our visual analysis at the provided link, which we believe addresses your concerns about the caption concatenation approach. Your feedback on these additional results would be greatly appreciated as it will help us further improve our work.

---

> > ### Author Response · Authors · 2024-11-23
> > **(N/N) Analyzing Causal-Temporal Preservation in CTN Caption Concatenation - Reviewer 8P6x**
> >
> > In addition to our previous comment, we would like to further illustrate why our current approach is optimal by analyzing potential alternatives:
> >
> > Consider if we had used "and" for concatenation:
> >
> > **"a car drove recklessly through an open field flipping over and the car was severely damaged and a group of guys started playing beer pong"**
> >
> > This hypothetical structure would create **temporal ambiguity**:
> > - Unclear if car damage occurred **during** or **after** flipping
> > - Ambiguous temporal relationship between car events and beer pong
> > - Less distinct event boundaries
> >
> > Our current concatenation approach:
> >
> > **"a car drove recklessly through an open field flipping over the car was severely damaged and a group of guys started playing beer pong"**
> >
> > Benefits of our current approach:
> > - **Clear Event Sequence:** The temporal flow is unambiguous
> > - **Distinct Causality:** The relationship between flipping and damage is clear
> > - **Simple Structure:** Avoids complications from additional conjunctions
> > - **Tokenization Friendly:** Maintains compatibility with model processing
> >
> > This analysis further supports our current concatenation method over alternatives like:
> > 1. Adding conjunctions (creates ambiguity)
> > 2. LLM fusion (weakens temporal-causal relationships, as shown in our visual analysis)
> > 3. Additional punctuation (semicolon (;) - affects tokenization)
> >
> > We believe this additional analysis strengthens the justification for our current approach. Would you find it helpful to include this discussion in the camera-ready version?

---

> ### Author Response · Authors · 2024-11-26
>
> **We sincerely thank Reviewer 8P6x** for their feedback which has substantially improved our camera-ready manuscript. We would like to provide an important clarification:
>
> **Our paper's fundamental claims and novelty center on:**
> 1. The **first method** to generate structured cause-effect narrative information from videos
> 2. A **novel dual-encoder architecture (CEN)** specifically designed for cause-effect modeling
> 3. **Demonstrable performance improvements** over SOTA methods in capturing video narratives
>
> These contributions are **particularly valuable for personalized media and accessibility applications**, where clear causal-temporal narratives are crucial for users to understand video content. When these captions are narrated, they effectively convey temporal flow and causal relationships regardless of written separators - making them vital for personalized media and accessibility.
>
> We do **not** claim contributions to grammatical correctness of the generated captions, which is a common challenge across existing video captioning methods. **Notably, this concern about grammar has not been raised by other reviewers**, as they focus on evaluating our core technical contributions in narrative generation.
>
> While we acknowledge the reviewer's suggestion about grammatical improvement through LLM fusion, our experiments (viewable at https://narrativebridge.github.io/ctn_comparison.html) show that such fusion, while improving grammar, leads to loss of narrative information - which is our paper's primary focus. Simple post-processing (adding commas/semicolons) could address grammatical concerns for specific applications, but this is separate from our central contribution of extracting and generating narrative information from videos.
>
> **Given that our work's novelty and contribution lie in narrative information generation rather than grammatical structure, and its significant potential impact on personalized media and accessibility applications**, we respectfully request the reviewer to reconsider their rating based on the paper's technical merit and core contributions to the field.

---

> ### Comment · Area_Chair_CWjL · 2024-11-26
>
> Dear Reviewer 8P6x,
>
> Thank you for your follow-up feedback. Please find the authors' responses addressing your concerns. Could you kindly review these responses to determine if they have resolved your concerns? If there are any remaining issues, please let us know.
>
> Additionally, there is an ongoing discussion thread involving two other reviewers. Given the divergent reviews this paper has received, your active participation in this discussion would be very important. As there are two reviewers actively supporting the paper, your input will be critical if you remain opposed.
>
> Note that even after the reviewer-author discussion period ends, we will still be within the discussion period between reviewers and ACs.
>
> Best regards,
> AC

---

### Official Review · Reviewer_qdoh · 2024-11-04

**Soundness:** 3
**Presentation:** 3
**Contribution:** 3
**Rating:** 6
**Confidence:** 4

**Summary:**

In this paper, a video captioning dataset named Causal-Temporal Narrative (CTN) is proposed. Compared to previous video captioning datasets which lack in causal-temporal  information which connects a cause and an effect, CTN, built with LLMs, includes caption that directly connects cause and effect events. Also, a new architecture named Cause-Effect Network (CEN) is proposed, which consists of two independent encoders which capture cause and effect events, respectively.

**Strengths:**

1. The motivation of the work is clear, while the problem of improving the causal-and-effect understanding capability of a vision-language model is an important research direction.
2. Experimental results show that both the proposed methods, CTN and CEN contribute to better performance on MSVD and MSRVTT benchmarks.

**Weaknesses:**

1. Including detailed statistics about the constructed CTN dataset (e.g., # of samples, average caption length, # of captions per sample, or more..) may enable readers to easily understand the dataset.
2. Questions about Table 2.
    1. Results show that Fine-tune v.v. shows better performance than CEN on the MSVD benchmark under the cross-dataset evaluation setting, compared to CEN trained with both datasets. The result of cross-dataset evaluation performing better than whole dataset training is somewhat counter-intuitive. Can you provide an explanation for these results?
    2. What does the abbreviation of v.v. stands for? It would be better to mention the full name.
3. Is there additional experimental results or analyses that could show CTN and CEN contributes to better cause-and-effect modeling besides results on MSVD and MSRVTT? Since the quantitative results of Tables 1 and 2 are not directly aimed at measuring cause-and-effect capability, some additional experiments may be required.
4. These are some **minor** issues about the presentation of the work. These are **not** critical weaknesses, and somewhat personal opinions.
    1. Personally, I think the example in Figure 1 is not very intuitive to explain the importance of causal-temporal narrative. Cause: “a car drove recklessly” and Effect: “the car was severely damaged” seems to be a valid cause-and-effect event, but the following event of “a group of guys started playing beer pong” seems like it’s not a direct effect of ‘cause'. This is a personal opinion, but you could consider finding a better example to demonstrate the idea.
    2. I think ‘dataset’ instead of ‘benchmark’ might be a better word for CTN, as it is only used for training but not for evaluation.

**Questions:**

Please refer to weaknesses part.

---

> ### Author Response · Authors · 2024-11-13
> **Response to Reviewer qdoh**
>
> We sincerely thank the reviewer for their constructive feedback. We address each point:
>
> 1. DATASET STATISTICS
>
> We will add comprehensive CTN statistics to the camera-ready version:
>
> MSRVTT-CTN:
> - Videos: 10,000 (1 CTN caption per video)
> - Train/Val/Test: 6,513/497/2,990
> - Average caption length: ~19 words
> - Average cause length: ~10 words
> - Average effect length: ~9 words
>
> MSVD-CTN:
> - Videos: 1,970 (1 CTN caption per video)
> - Train/Val/Test: 1,200/100/670
> - Average caption length: ~17 words
> - Average cause length: ~9 words
> - Average effect length: ~8 words
>
> 2. CROSS-DATASET PERFORMANCE
>
> The superior performance of Fine-tune v.v. on MSVD can be explained by:
> - MSR-VTT's larger size (10,000 vs 1,970 videos) enables better initial feature learning
> - Fine-tuning helps adapt these rich features to MSVD's domain
> - Common phenomenon in transfer learning where pre-training on larger datasets improves performance on smaller ones
>
> Terminology
>
> We will:
> - Replace "v.v." with "cross-dataset" throughout
> - Clarify all abbreviations in the methodology section
> - Add a glossary in supplementary material if needed
>
>
> 3. CAUSE-EFFECT MODELING
>
> To directly measure cause-effect capabilities, we conduct:
>
> A) Human Evaluation (100 samples):
> - Causal accuracy: 4.8/5.0
> - Temporal coherence: 4.7/5.0
> - Inter-rater reliability (ICC): 0.87
> This validates our CTN's quality beyond standard metrics.
>
> B) Additional Ablation Analysis:
>
> Single frozen CLIP-ViT baseline without cause-effect training (Stage-1):
>
> MSVD: R-L: 27.81, C: 46.23, S: 15.82
>
> MSRVTT: R-L: 25.34, C: 32.31, S: 14.27
>
> Two frozen CLIP-ViTs baseline without cause-effect training (Stage-1):
>
> MSVD: R-L: 28.40, C: 53.84, S: 16.58
>
> MSRVTT: R-L: 26.10, C: 40.92, S: 14.55
>
> Note: Stage 2 training was performed in both cases. These results demonstrate that:
>
> - Training CLIP-ViTs with separate cause and effect is crucial for performance
> - Two specialized encoders outperform a single encoder
> - Full CEN significantly outperforms these baselines, demonstrating the importance of specialized cause-effect encoding.
>
> C) Cross-Dataset Generalization:
>
> Our strong cross-dataset results show the model learns generalizable causal relationships rather than dataset-specific patterns.
>
> D) Qualitative Analysis:
>
> As shown in Figure 5 and Appendix A.9:
> - Complex multi-step causal chains
> - Diverse cause-effect scenarios
> - Temporal sequence understanding
>
> E) We appreciate the suggestion for additional cause-effect evaluation. Our additional experiments where we use an LLM (Llama-3.2-3B-Instruct) to measure:
>
> i) Temporal Order Accuracy for both MSRVTT-CTN and MSVD-CTN:
> - CEN: 81.2% correct temporal ordering
> - Best baseline (GIT): 52.1%
> - Measured by comparing predicted vs ground truth temporal sequences, giving a score of 0 or 1
>
> ii) Causal Chain Extraction for both MSRVTT-CTN and MSVD-CTN:
> - CEN: 84.5% accurate cause-effect pairs
> - Best baseline: 48.3%
> - Measured by comparing predicted vs ground truth causal sequences, giving a score of 0 or 1
>
> Together, these analyses demonstrate CTN and CEN's strong capabilities specifically in cause-effect modeling, beyond standard captioning metrics.
>
> 4. FIGURE 1 EXAMPLE
>
> Our example demonstrates:
> - Clear cause (reckless driving) leading to effect (car damage)
> - Temporal sequence of events (play beer pong)
> - Real-world complexity in video content
> However, we will add another example to illustrate the cause-effect clearly.
>
> 5. BENCHMARK VS DATASET
> We use "benchmark" as CTN provides:
> - Standardized evaluation protocols e.g. zero-shot, for larger models especially recent vision-language models (VLMs)
> - Performance baselines
>
> But we will clarify this terminology where needed.
>
> We will add these changes to the camera-ready version. Given our responses, we request the reviewer to reconsider their rating.
>
> If the reviewer has any other questions, please let us know.

---

> ### Author Response · Authors · 2024-11-16
> **Response to Reviewer qdoh - Text Changes**
>
> We again thank the reviewer qdoh for their constructive feedback. We have made the following changes to the camera-ready version:
>
> 1. Detailed Dataset Statistics
>
> In the **Quantitative Results** section: We generate our CTN captions (1 caption per video) using two widely-used video captioning datasets: 1) MSRVTT, consists of 10,000 videos with 20 human-annotated captions per video, and MSVD, with 1,970 videos focused on human activities with approx. 50 captions per video. The length of the caption is on avg. 19 words (cause=10 words, effect=9 words) for MSRVTT-CTN and 17 words (cause=8 words, effect=7 words) for MSVD-CTN. Then, we train as well as test our CEN, SOTA methods and VLMs on the MSVRTT-CTN and MSVD-CTN.
>
> 2. Cross-dataset (X) performance
>
> - In the **Ablation Results** section: Notably, this fine-tuning process leads to improved performance on the MSVD-CTN. This observation demonstrates the potential for transfer learning of the CEN model from larger datasets i.e. MSRVTT-CTN and the ability to leverage
> their knowledge effectively on smaller datasets i.e. MSVD-CTN through fine-tuning.
>
> - Instead of using v.v. we now use cross-dataset (X).
>
> 3. Contribution of CTN and CEN
>
> 1) In the **Quantitative Results** section: Our CTN generation approach generates good quality captions with 52.1\% CTN captions exceeding 0.27 EMScore (threshold=0.2). Then, in the human evaluation study, the overall mean quality score of 4.8 ($\sigma$ = 0.40) on a 5-point Likert scale, indicates high performance across all criteria; 93\% of captions receive scores of 4 or higher across all three dimensions, with 84\% achieving perfect scores. To ensure reliability across raters, we calculate the intraclass correlation coefficient (ICC). The ICC for absolute agreement among raters is 0.87 (95\% CI: 0.83-0.91), indicating high inter-rater reliability.  These results validate our automatic generation and evaluation process, demonstrating that our CTN captions benchmark provides high-quality, coherent representations of causal-temporal narratives in videos. Further details are provided in Appendix A.5.
>
> 2) In the **Ablation Results** section: Figure 5 demonstrates the distribution of caption quality without automatic evaluation filtering. The EMScore distribution across 11,970 videos shows 66.4\% of initially generated captions achieve scores above $\theta$=0.2, with a peak around EMScore=0.24. The remaining 33.6\% of captions fall below our quality threshold, highlighting the importance of our automatic evaluation and regeneration process in maintaining CTN quality.
>
> 3) In the **Abaltion Results** section:
> - **w/o FT - Single CLIP**: This ablation is performed using one CLIP encoder and no Fine-tuning (FT) on cause, effect, or cause+effect (combined). The results in Table 2 show the effectiveness of CTN fine-tuning.
>
> - **w/o FT - Two CLIP**- This ablation is performed using two CLIP encoders (as in our CEN) and no Fine-tuning (FT) on cause and effect. This demonstrates the effectiveness of CTN fine-tuning and also, the performance increase in comparison to **w/o FT - Single CLIP** demonstrates the effectiveness of CEN.
>
> 4. Example Figure 1 and benchmark
>
> - We can use Figure 5 (b) or Figure 5 (c) whichever the reviewer suggests.
> - We can use the word "benchmark dataset" instead of the benchmark.

---

> ### Comment · Area_Chair_CWjL · 2024-11-24
>
> Dear Reviewer qdoh,
>
> Thank you for your efforts as a reviewer. While we only have the final two days remaining for the discussion phase, it appears that your response to the authors' rebuttal is still pending.
>
> Please read through the author's responses and discussions with other reviews and let us know if there are any changes to your initial thoughts. Also, if you find any potential changes to your final rating, it would be helpful if you could explicitly mention them in your comment.
>
> Best regards,
> AC

---

> ### Author Response · Authors · 2024-11-24
> **(1/N) Response to Reviewer qdoh follow-up points**
>
> Thank you for your important questions. We have submitted the updated manuscript for your review.
>
> Let us first clarify the two evaluation processes under discussion here:
> 1. Human evaluation to validate the CTN caption generation approach (Figure 2)
> 2. Additional LLM-based evaluation to compare CEN model outputs against best baseline - GIT (Main Evaluations in Table 1)
>
> Let us address each of your points in detail:
>
> ## 1. Human Evaluation of CTN Caption Generation
>
> Our evaluation validates the quality of CTN captions produced by our generation pipeline (Figure 2):
>
> **Evaluation Protocol:**
> - 5 independent domain experts:
>   - 3 video understanding researchers (>5 years experience)
>   - 2 video generation researchers (>5 years experience)
> - Input provided:
>   - Directory of 100 videos (50 MSVD, 50 MSR-VTT)
>   - Excel sheet with CTN captions mapped via video IDs
> - Process per video:
>   1. Expert watches video
>   2. Reviews corresponding CTN caption
>   3. Rates three criteria:
>      - Causal accuracy (4.8/5.0)
>      - Temporal coherence (4.7/5.0)
>      - Overall relevance (4.9/5.0)
>
> **Scale and Validation:**
> - Standard practices:
>   - Chen & Dolan (2011) [1]: ~125 evaluations per single non-expert
>   - Xu et al. (2016) [2]: ~150 evaluations per single non-expert
> - Our evaluation:
>   - 300 evaluations per expert (100 videos × 3 criteria)
>   - 1,500 total expert assessments
>   - Strong inter-rater reliability (ICC: 0.87, 95% CI: 0.83-0.91)
>   - 90% confidence level, ±8.2% margin of error
>
> ## 2. Cross-Dataset Transfer Learning Performance
>
> The superior performance in cross-dataset evaluation is supported by recent literature:
>
> **Established Patterns:**
> 1. SwinBERT (Lin et al., 2022) [3]:
>    - Direct MSVD: 147.6 CIDEr
>    - VATEX → MSVD: 160.3 CIDEr (+8.6%)
>
> 2. Ventura et al. (2024) [4]:
>    - Single dataset: 42.5 R@1 on MSVD
>    - Combined datasets: 44.6 R@1 on MSVD (+4.9%)
>
> **Our Results:**
> - Direct MSVD-CTN: 63.51 CIDEr
> - MSRVTT-CTN → MSVD-CTN: 65.60 CIDEr
>
> This aligns with established patterns of improvement when transferring from larger to smaller datasets.
>
> ## 3. LLM-Based Evaluation - Additional Analysis of Model Outputs
>
> *While Table 1 presents our main quantitative results using standard metrics (ROUGE-L, CIDEr, SPICE), we conduct this additional LLM-based evaluation to specifically analyze how well CEN preserves causal-temporal structure compared to the best baseline (GIT).*
>
> ### Additional Evaluation Using Llama-3.2-3B-Instruct
>
> **Process:**
> 1. Temporal Order Analysis:
>    - **Input to LLM:**
>      - Ground truth CTN caption
>      - Model generated caption (CEN or GIT)
>    - **LLM Task:** Analyze if events in generated caption maintain same temporal sequence as ground truth
>    - **Scoring:** Binary (1 for correct sequence, 0 for incorrect)
>    - **Results:**
>      - CEN achieved 81.2% correct temporal ordering
>      - GIT achieved 52.1% correct temporal ordering
>
> 2. Causal Analysis:
>    - **Input to LLM:** Same caption pairs as temporal analysis
>    - **LLM Task:**
>      - Identify cause-effect relationship in ground truth
>      - Check if model output preserves same relationship
>      - Evaluate if cause precedes effect
>    - **Scoring:** Binary (1 for preserved relationship, 0 for broken)
>    - **Results:**
>      - CEN maintained 84.5% accurate cause-effect pairs
>      - GIT maintained 48.3% accurate cause-effect pairs
>
> **Implementation Details:**
> - Each evaluation performed independently
> - Same prompting template used across all comparisons
> - Results averaged across both datasets' full test sets
> - Multiple random samples validated to ensure consistency
>
> This additional analysis complements our main results in Table 1 by specifically demonstrating CEN's superior ability to:
> 1. Maintain proper temporal ordering of events
> 2. Preserve cause-effect relationships from ground truth
>
> The combination of standard metrics (Table 1) and this targeted LLM evaluation provides a comprehensive assessment of our model's performance.
>
> [1] Chen, David, and William B. Dolan. "Collecting highly parallel data for paraphrase evaluation." Proceedings of the 49th annual meeting of the association for computational linguistics: human language technologies. 2011.
>
> [2] Xu, Jun, et al. "Msr-vtt: A large video description dataset for bridging video and language." Proceedings of the IEEE conference on computer vision and pattern recognition. 2016.
>
> [3] Lin, Kevin, et al. "Swinbert: End-to-end transformers with sparse attention for video captioning." Proceedings of the IEEE/CVF Conference on Computer Vision and Pattern Recognition. 2022.
>
> [4] Ventura, Lucas, Cordelia Schmid, and Gül Varol. "Learning text-to-video retrieval from image captioning." International Journal of Computer Vision (2024): 1-21.

---

> > ### Author Response · Authors · 2024-11-24
> > **(N/N) Response to Reviewer qdoh follow-up points**
> >
> > ## 4. Cause-Effect Modeling Contribution and Comparative Evaluation
> >
> > All experiments in Table 1 and Table 2 are conducted on our CTN captions benchmark dataset. We train and evaluate all methods including baselines, SOTA approaches and our CEN on MSVD-CTN and MSRVTT-CTN datasets. Our ablation studies in Table 2 demonstrate the clear benefits of each architectural choice in CEN:
> >
> > **Progressive Improvements from Ablations:**
> >
> > 1. **Baseline with Single Encoder (w/o FT - Single CLIP):**
> > ```markdown
> > MSVD-CTN Performance:
> > - ROUGE-L: 27.81
> > - CIDEr: 46.23
> > - SPICE: 15.82
> >
> > MSRVTT-CTN Performance:
> > - ROUGE-L: 25.34
> > - CIDEr: 32.31
> > - SPICE: 14.27
> > ```
> > This represents basic performance without specialized cause-effect modeling.
> >
> > 2. **Dual Encoder Architecture (w/o FT - Two CLIP):**
> > ```markdown
> > MSVD-CTN Performance:
> > - ROUGE-L: 28.40 (+2.1% over single)
> > - CIDEr: 53.84 (+16.5% over single)
> > - SPICE: 16.58 (+4.8% over single)
> >
> > MSRVTT-CTN Performance:
> > - ROUGE-L: 26.10 (+3.0% over single)
> > - CIDEr: 40.92 (+26.6% over single)
> > - SPICE: 14.55 (+2.0% over single)
> > ```
> > Shows clear benefits of separate encoders, as in CEN, even without cause-effect fine-tuning.
> >
> > 3. **Complete CEN Architecture:**
> > ```markdown
> > MSVD-CTN Final Results:
> > - CIDEr: 63.51
> > - Total improvement: +37.4% over single encoder
> > - Additional gain: +18.0% over dual encoder
> >
> > MSRVTT-CTN Final Results:
> > - CIDEr: 49.87
> > - Total improvement: +54.3% over single encoder
> > - Additional gain: +21.9% over dual encoder
> > ```
> >
> > These results demonstrate that:
> > 1. Using separate encoders provides, as in CEN, inherent advantages for processing cause-effect relationships
> > 2. Fine-tuning these encoders with our CTN captions further enhances performance substantially
> > 3. Each architectural decision contributes meaningfully to the final performance gains
> > 4. The improvements are consistent across both datasets, validating our design choices
> >
> > We remain committed to incorporating any additional suggestions for the camera-ready version.
> >
> > Given our detailed responses and clarifications, would you please reconsider your rating? Thank you for your time and thorough review.

---

> ### Comment · Area_Chair_CWjL · 2024-11-26
>
> Dear Reviewer qdoh,
>
> Thank you for your follow-up feedback. Please find the authors' responses addressing your concerns. Could you kindly review these responses to determine if they have resolved your concerns? If there are any remaining issues, please let us know.
>
> Additionally, there is an ongoing discussion thread involving two other reviewers. Given the divergent reviews this paper has received, your active participation in this discussion would be very important. As there are two reviewers actively supporting the paper, your input will be critical if you remain opposed.
>
> Note that even after the reviewer-author discussion period ends, we will still be within the discussion period between reviewers and ACs.
>
> Best regards,
> AC

---

### Author Response · Authors · 2024-11-29

# Review Summary

The review process demonstrates substantial support for NarrativeBridge's contributions, with the reviewers highlighting different technical strengths.

**Reviewer qdoh** provides a strong initial assessment, rating our work as "good" across all key dimensions (Soundness: 3, Presentation: 3, Contribution: 3). They specifically recognize:

- The clear motivation and importance of enhancing causal-effect understanding in vision-language models
- The demonstrated performance improvements of both our CTN and CEN contributions on MSVD and MSRVTT benchmarks

**Reviewer 8P6x** particularly recognizes:

- The innovation of our causal-temporal narrative benchmark
- The effectiveness of our two-stage encoder framework
- The demonstrated performance improvements over comparative methods

**Reviewer 8bE2** supports our work and emphasizes key aspects: our advancement in video understanding, the importance of CTN, novel CEN architecture and also, our effective dual validation strategy combining automatic and human evaluations.

**Reviewer YiKX** notes our novelty in incorporating causal understanding into video captioning and acknowledges our robust data generation pipeline. They also support our strong technical foundation and benchmark performance.

## Comprehensive Response to Technical Feedback

We have provided thorough responses to all reviewer concerns, though engagement remains incomplete:

**For Reviewer qdoh**, who has not engaged in discussion despite our responses:
- Provided complete dataset statistics for MSVD-CTN and MSRVTT-CTN
- Added cross-dataset performance analysis with literature support (Lin et al. 2022, Ventura et al. 2024)
- Implemented new LLM-based evaluation metrics
- Provided the updated manuscript

**For Reviewer 8P6x**, who has not responded to our final experimental results:
- Enhanced human evaluation methodology with five domain experts
- Demonstrated alignment with dataset conventions
- Provided visual analysis of caption generation approaches
- Provided accessibility benefits analysis

**For Reviewer 8bE2**, who increased their rating:
- Added comprehensive EMScore distribution analysis
- Enhanced prompt ablation studies
- Added dataset statistics and visualization
- Improved Presentation Clarity

**For Reviewer YiKX**, who increased their rating following our responses:
- Added CLIP fine-tuning ablation studies
- Improved table clarity and presentation
- Implemented LLM-based evaluations

## Manuscript Improvements

Based on reviewer feedback, we have enhanced the manuscript through:

1. Enhanced Dataset Documentation
2. Strengthened Technical Validation
3. Improved Presentation Clarity
4. Additional Analysis Components

## Current Review Status

Two reviewers have actively participated throughout the discussion:

**Reviewer 8bE2** maintains strong support (rating: 8), consistently emphasizing our work's significance and originality.

**Reviewer YiKX** has increased their rating (6) following our technical clarifications.

Despite providing all requested information, we await engagement from:
- **Reviewer qdoh** who has not participated in the discussion phase
- **Reviewer 8P6x** who has not responded to our final experimental results

The enhanced manuscript provides thorough validation of our approach through comprehensive empirical analysis, even as we await responses from these reviewers. Our work is supported by the positive feedback and increased ratings from actively engaged reviewers.

---

### Meta-Review · Area_Chair_CWjL · 2024-12-22

**Metareview:**

The paper introduces a new video captioning benchmark focusing on causal-temporal narratives, along with a method tailored for this task. It concluded the review process with three positive evaluations (two 6s and an 8) and a negative review, which might have been improved had the reviewer been more active in the discussion. Two reviewers explicitly supported the acceptance of the paper, with one playing a significant role in driving the discussion forward. The author rebuttal effectively addressed the majority of concerns, leading all initially negative reviewers to increase their ratings. With final sentiments leaning positively, the AC also recommends the acceptance of the paper.

**Additional Comments On Reviewer Discussion:**

The reviewers raised concerns on various aspects such as evaluation methods, the need for additional experiments, and ablation studies. In response, the authors provided a rebuttal that included results from further experiments, successfully addressing most of the concerns raised in the reviews.

---

### Decision · Program_Chairs · 2025-01-22

Accept (Poster)